**Subject Category:**
Biology (whole organism)

microbiology/molecular biology/plant science

agarwood, *Aquilaria sinensis*, fungal community, fungal diversity, two-dimensional gas chromatography with high-resolution time-of-flight mass spectrometry, volatile compound

**Authors for correspondence:**
Yuan Yuan
e-mail: y_yuan0732@163.com
Luqi Huang
e-mail: huangluqi01@126.com

[†]These authors contributed equally to this study.

# Agarwood wound locations provide insight into the association between fungal diversity and volatile compounds in *Aquilaria sinensis*

Juan Liu[1,†], Xiang Zhang[2,†], Jian Yang[1], Junhui Zhou[1], Yuan Yuan[1], Chao Jiang[1], Xiulian Chi[1] and Luqi Huang[1,2]

[1]National Resource Center for Chinese Materia Medica, China Academy of Chinese Medical Sciences, Beijing 100700, People's Republic of China
[2]Department of Traditional Chinese Medicine, Guangdong Pharmaceutical University, Guangzhou 510006, People's Republic of China

(iD) JL, 0000-0002-5319-6065

The aim of the present study was to investigate the effect of wound location on the fungal communities and volatile distribution of agarwood in *Aquilaria sinensis*. Two-dimensional gas chromatography with high-resolution time-of-flight mass spectrometry revealed 60 compounds from the NIST library, including 25 sesquiterpenes, seven monoterpenes, two diterpenes, nine aromatics, nine alkanes and eight others. Of five agarwood types, Types IV and II contained the greatest number and concentration of sesquiterpenes, respectively. The fungal communities of the agarwood were dominated by the phylum Ascomycota and were significantly affected by the type of wound tissue. Community richness indices (observed species, Chao1, PD whole tree, ACE indices) indicated that Types I and IV harboured the most and least species-rich fungal communities, and the fungal communities of Types V, I, III and IV/II were dominated by *Lasiodiplodia*, *Hydnellum*, *Phaeoisaria* and *Ophiocordyceps* species, respectively. Correlations between fungal species and agarwood components revealed that the chemical properties of *A. sinensis* were associated with fungal diversity. More specifically, the dominant fungal genera of Types V, I and III (*Lasiodiplodia*, *Hydnellum* and *Phaeoisaria*, respectively) were strongly correlated with specific terpenoid compounds. The

finding that wound location affects the fungal communities and volatile distribution of agarwood provides insight into the formation of distinct agarwood types.

# 1. Introduction

Agarwood, also known as 'wood of the Gods', is of huge cultural significance, due to its peculiar perfume and use in incense ceremonies. Derived from the resinous portions of trunks and branches from *Aquilaria* and *Gyrinops* species, it is the basis of some of the most world's most exclusive perfumes and is used extensively for medicine and incense across Asia, the Middle East and Europe [1,2]. This widespread use can likely be attributed to the sesquiterpene and phenylethyl chromone derivatives of agarwood [3–5], which have biological and pharmacological properties, including antimicrobial, anti-oxidant and anti-proliferative activities [6–8]. Moreover, agarwood has the potential to prevent cancer and to treat both gastric ulcers and cough in asthma patients [8–10]. Indeed, in China, agarwood, which is mainly derived from *Aquilaria sinensis* (Lour.) Spreng, is valued as a traditional Chinese medicine for the treatment of emesia, asthma and insomnia [11,12]. However, wild sources of agarwood are at serious risk of depletion, owing to the slow and infrequent formation of agarwood and to the resource's uncontrolled collection in forests. As a result, all *Aquilaria* and *Gyrinops* species are endangered and are listed in Appendix II of the Convention on International Trade in Endangered Species of Wild Fauna and Flora (CITES, http://www.cites.org). To protect both wild agarwood resources and sustainable agarwood production, *A. sinensis* has been widely cultivated in Guangdong and Hainan provinces in China, and due to its rarity and value, *A. sinensis* has also been used to investigate processes for improving the oil yield of agarwood in China.

Under natural conditions, agarwood is produced when attacked by microbes, insects or other damaging organisms and promotes the accumulation of agarwood resin [13]. Fungal infection has long been recognized as the cause of agarwood formation in *Aquilaria* host trees [14–16]; a variety of fungi (e.g. *Fusarium* sp., *Chaetomium globosum*, *Menanotus flavolives*, *Lasiodiplodia theobromae* and *Rigidoporus vinctus*) isolated from infectious *Aquilaria* trees have been reported to accelerate agarwood formation and to promote the accumulation of volatile compounds [2,17–21]. Agarwood is categorized, in China, into different types on the basis of the location at which it forms [22,23]. However, even though the various agarwood types are widely traded on the Chinese market, the relationship between the volatile compounds and fungal diversity of agarwood remains relatively unclear. Elucidating such relationships would help to delineate the numerous potential fungal niches and chemical characteristics of agarwood formed in different regions of the trees.

Using *A. sinensis* as a model system for agarwood formation, the aim of the present study was to investigate the variation of fungal communities across habitats within a tree host from different agarwood-formed tissues. Comprehensive two-dimensional gas chromatography with high-resolution time-of-flight mass spectrometry (GC × GC-HR-TOF-MS), which is a versatile analytical tool combining two powerful analytical technologies with complementary attributes [24,25], was also used to elucidate the volatile constituents of different types of agarwood. In addition, the regular patterns of fungal communities and volatile constituents observed in different types of agarwood facilitated the characterization of fungal community structures within different parts of agarwood formation, as well as the investigation of the relationship between fungal diversity and volatile compound production, which could help elucidate the variation of agarwood fragrance.

# 2. Material and Methods

## 2.1. Study location and sampling methods

Thirty-year-old *A. sinensis* trees in Dalingshan town, Guangdong province, China (latitude 22°45′43″ N, longitude 113°48′45″ E), were physically wounded using a machete according to the previous treated method [26,27], and after five years, 0.5- to 1-cm thick agarwood sections had formed beneath the wounded surface. In November 2017, the agarwood was collected by cutting at about 5 cm below the wound area, and the non-agarwood parts were removed from the samples. These trees were similar in diameter, about 20–30 cm and spaced at intervals of about 5–10 m. Fifteen trees were wounded at different locations as following, with each treatment group including three individual trees. The five types

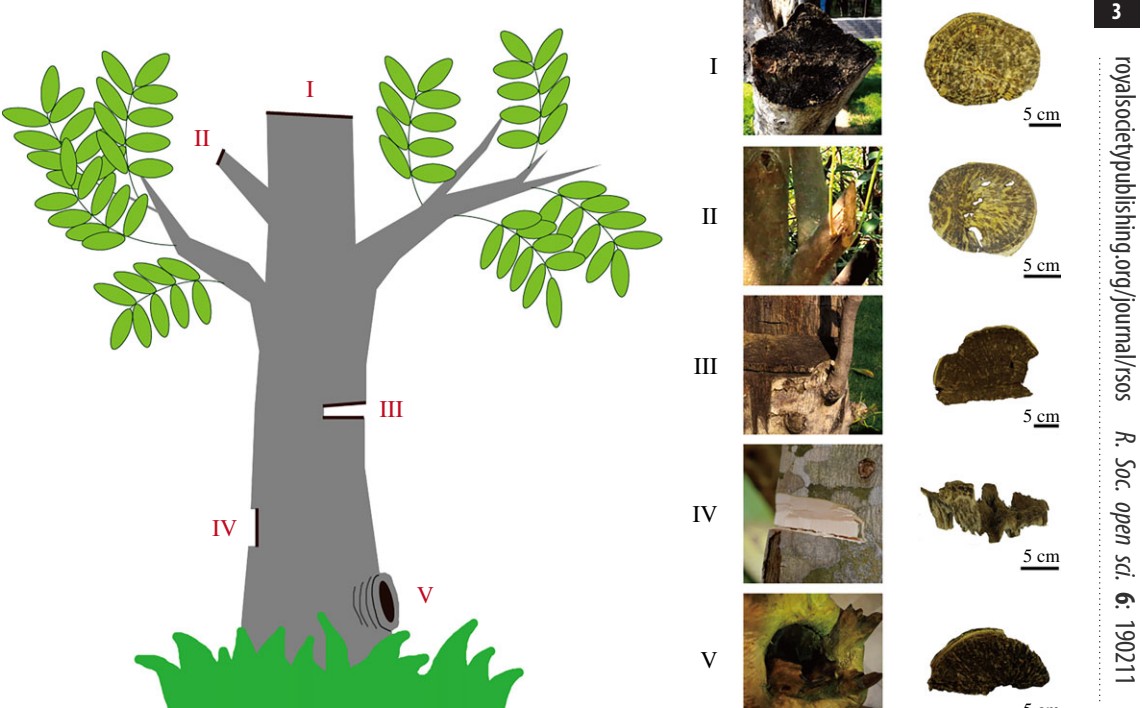

**Figure 1.** Wound tissues and agarwood sample collection from *Aquilaria sinensis*.

of agarwood that formed at the different wound points were considered Type I, II, III, IV and V, respectively (figure 1). Type I agarwood formed in the transverse section of wounded main trunks at about 2.0–2.5 m above the ground. Type II agarwood formed in the transverse section of wounded lateral branches at about 1.5–1.8 m above the ground. Type III agarwood formed in the transverse section of wounded main trunks at about 1.0–1.2 m above the ground. Type IV formed in the longitudinal section of main trunks at about 0.8–1.2 m above the ground, and Type V usually formed in the main trunks near the root, which was encapsulated by surrounding tissue (at about 0.2–0.4 m above the ground).

## 2.2. DNA extraction, PCR amplification and DNA sequencing

All agarwood samples were washed with distilled sterile water and were surface sterilized by soaking in 75% ethanol for 2 min, after which they were rinsed with sterile water and dried on sterile filter paper. Total DNA was extracted from each agarwood sample (100 mg) using the MoBio PowerPlant Pro DNA Isolation Kit (MoBio Laboratories, Inc., Carlsbad, CA, USA). Two replicate extractions were performed for each agarwood sample, in order to achieve sufficient DNA yields, and DNA quantity was determined using electrophoresis on 1% agarose gels. The extracted DNA was quantified using a NanoDrop 1000 spectrophotometer (NanoDrop Products, Wilmington, DE, USA). According to the concentration, each DNA sample was diluted to 1 ng µl$^{-1}$ using sterile water. Internal transcribed spacer 1 (ITS1) sequences were then amplified from all the samples using the universal primers ITS5-1737F (5′-GGAAGTAAAAGTCGTAACAAGG-3′) and ITS2-2043R (5′-GCTGCGTTCTTCATCG ATGC-3′) and barcodes to distinguish between the samples. The PCR was performed using a Phusion High-Fidelity PCR Master Mix (New England Biolabs, Ipswich, MA, USA), according to Cregger *et al.* [28]. The resulting PCR products were mixed in equal ratios and purified using a Qiagen Gel Extraction Kit (Qiagen, Duesseldorf, Germany). Sequencing libraries were then generated using the Ion Plus Fragment Library Kit (Thermo Fisher Scientific, Waltham, MA, USA), following the manufacturer's recommendations, and index codes were added. Library quality was assessed using a Qubit v. 2.0 Fluorometer (Thermo Fisher Scientific) and an Agilent Bioanalyzer 2100 system (Agilent Technologies, Inc., Palo Alto, CA, USA), and the library was sequenced using a Thermofisher Life Ion S5 platform (Thermo Fisher Scientific). The high-throughput sequencing data are available from the National Center for Biotechnology Information (NCBI) database in the Sequence Read Archive (SRA) database, under BioProject number PRJNA509099.

## 2.3. Bioinformatics processing

Raw sequences ($2 \times 300$-bp reads) were generated using an Illumina MiSeq system (Illumina, San Diego, CA, USA) and MiSeq reagent kit v. 3 (Illumina) by the University of Wisconsin Biotechnology Center (Madison, WI, USA) [29]. Low-quality reads were first removed using Cutadapt v. 1.9.1 (http://cutadapt.readthedocs.io/en/stable/), and paired-end reads were categorized according to the unique barcodes that were then removed along with the primer sequences. Overlapping reads were merged using FLASH [30], and high-quality clean tags were filtered using the QIIME quality-control process [31]. The tags were compared with the Unite database [32], using the UCHIME algorithm to remove chimaeric sequences and to obtain clean reads [33]. High-quality sequences were clustered into operational taxonomic units (OTUs), which were defined as 97% similar, using the UPARSE software [34]. These OTUs were applied to analyse diversity, richness and rarefaction curves using MOTHUR [35]. Taxonomic assignments of OTUs that achieved 97% similarity were obtained using the QIIME v. 1.9.1 [31] software package through comparison with the SILVA [36], Greengene [37] and RDP [38] databases. Subsequent analyses on alpha and beta diversity were performed based on the normalized OTU abundance. Venn diagrams, alpha diversity (including Chao1 and ACE richness estimators, Shannon and Simpson diversity indices, and phylogenetic diversity whole tree), beta diversity (including principal co-ordinates analysis, principal component analysis and non-metric multidimensional scaling) and heat map analysis were performed to identify mutual and unique taxa between groups, and the above analysis of fungal species diversity in samples using R software (http://www.r-project.org/).

## 2.4. Extraction of volatile oil

Dried agarwood powder (0.1 g) from each sample was separately weighed and placed in a 2-ml centrifuge tube, and 1.5 ml ethyl acetate was added to the tube. The powder was soaked overnight at room temperature and extracted for 45 min using the 40-kHz ultrasonic cold extraction method, according to Liao *et al.* [39] with slight modification. The solvent phase (upper layer) was separated by centrifugation at 12 000 r.p.m. and 4°C for 10 min. After adding ethyl acetate to supplement the reduced weight, the volatile oil was filtered through a 0.22-μm PTFE filter membrane and then stored in a non-transparent glass bottle at 4°C prior to GC × GC-HR-TOF-MS analysis.

## 2.5. GC × GC-HR-TOF-MS analysis

The GC × GC-HR-TOF-MS system consisted of an East & West 3300 GC × GC equipped with a TOF-MS (East & West Analytical Instrument Co., Beijing, China), which is used to acquire mass spectral data from the GC × GC. The first separation was performed in a conventional non-polar GC column Agilent DB-5MS (30 m × 0.25 mm inner diameter × 0.25 μm film thickness), and the second was performed in a medium GC column Agilent DB-17HT (2.5 m × 0.25 mm inner diameter × 0.15 μm film thickness). An aliquot (1 μl) of each sample was separately injected into the GC injector, using a split ratio of 30 : 1 at 300°C. The separation was performed under the following conditions: initial temperature of 70°C for 1 min, ramped at 6°C min$^{-1}$ to 300°C, and held at 300°C for 1 min. The transfer line into the TOF-MS source was heated to 270°C, and the electron impact ionization source was operated at 240°C with a collision energy of 70 eV and an acquisition voltage of 1800 V. The mass spectrometer was operated at an acquisition rate of 100 spectra s$^{-1}$, ranging from 50 to 550 u. The relative contents of the individual components of each sample were expressed in percentage of peak area relative to the total peak area. The identification of volatiles of the agarwood samples was based on a National Institute of Standards and Technology 11 (NIST11) library search combined with the Kovats retention index (RI) [40]. Compounds with lower search probabilities (less than 60) were regarded as unknowns. The mass spectral match factor (probability > 60) was used to judge whether a peak was correctly identified or not. For the determination of RI, calculated on the first dimension, a series of n-alkanes (C9–C23) was used under the same experimental conditions. The RI for each compound was calculated as follows:

$$\mathrm{RI} = 100 \times \left\{ n + \frac{\log t'(i) - \log t'(n)}{\log t'(n+1) - \log t'(n)} \right\},$$

where $n$ and $n+1$ are the number of carbon atoms in alkanes eluting before and after the compound, respectively; $t'(n)$ and $t'(n+1)$ were the corresponding retention time values and $t'(i)$ was the retention time of the identified compound.

## 2.6. Statistical analysis

The mean and standard error (s.e.) of all data were calculated, and all values were reported as the mean of three replicates. The community richness and diversity indices of the fungi and volatile contents of the five types of agarwood were analysed using one-way analysis of variance (ANOVA), with a significance level of $p < 0.05$. Heat map analysis was used to analyse the abundance of fungal distribution and the concentration of volatile compounds. Principal component analysis (PCA) was also performed to investigate the differences in volatile contents among the above agarwood samples. In addition, Pearson correlation analysis was used to investigate the relationship between fungal communities and volatile components. All statistical analyses were performed using SPSS v. 16.0 (SPSS, Inc., Armonk, NY, USA) [41].

# 3. Results

## 3.1. Identification of characterized agarwood components

To investigate the relationship between agarwood characteristics and the presence of specific fungal species, the volatile compounds of different agarwood types were first identified using GC × GC-HR-TOF-MS. More specifically, the volatile compounds of five different agarwood samples were separated and identified using a DB-5MS column on the first dimension and a DB-17HT column on the second dimension via GC × GC-HR-TOF-MS analysis. This analysis identified 60 strong matches to the NIST library data, including 25 sesquiterpenes, seven monoterpenes, two diterpenes, nine aromatics, nine alkanes and eight other components (table 1 and figure 2). The agarwood volatile fraction was characterized by a high percentage of sesquiterpenes, including guaianes (compounds 1–5; figure 2), eudesmanes (compounds 7–13; figure 2), agarofurans (compound 14; figure 2) and several others (compounds 6, 15–25; figure 2). Of these sesquiterpenes, two compounds, namely α-eudesmol and α-copaen-11-ol, were found in all the agarwood samples (electronic supplementary material, figure S1, and table 1).

## 3.2. Analyses of volatile compounds from various types of agarwood

Of the five agarwood types, Types IV and II contained the greatest number and concentration of distinct sesquiterpenes, respectively (table 1), and Types I and II contained greater monoterpene concentrations than Types IV and V. In addition, Type V had the greatest diterpene concentration, and Types I and III contained high levels of alkanes (table 1). A concentration heat map of volatile compounds was generated to further analyse the volatile profiles of the agarwood samples (figure 3). From this, Type II was clustered on a single branch of the cluster tree, which suggested that the volatile compounds of Type II were quite different from the others.

## 3.3. Structure and diversity of agarwood-associated fungal communities

Between 52 594 and 80 303 qualified reads were obtained from each of the five types of agarwood, and were attributed to six phyla, 25 classes, 63 orders, 115 families and 141 genera of fungi, with 221–378 OTUs identified with a similarity of 97% (table 2). The species diversity of Type IV was lower than that of the others (table 2), and the samples of each agarwood type possessed more common OTUs than specific ones (figure 4). Significantly more fungal species were observed in Type I than in Type IV ($p < 0.05$; figure 5a). Other community richness indices, including PD whole tree, ACE and Chao1, also indicated that the fungal community of Type I was richer than that of the other types and that the fungal community of Type IV possessed the lowest richness (figure 5b−d). However, the Shannon and Simpson indices identified Type III as possessing the greatest fungal diversity and Type II as possessing the lowest (figure 5e,f). UniFrac analysis, including PCoA, NMDS and PCA, indicated that Types III and I were distinct from Types II and IV (figure 6a−c).

## 3.4. Distribution of fungi among agarwood types

A heat map was generated to further analyse the taxonomic distribution of fungi among the different types of agarwood (figure 7). Among the five detected fungal phyla, the Ascomycota was the most dominant in all the agarwood samples, and Basidiomycota and Zygomycota were more abundant in

**Table 1.** Volatile compounds identified in the agarwood samples using GC×GC-HR-TOF-MS. Library probability of each compound was above 60 Mol. Wt., molecular weight. CAS No., library CAS number.

| no. | compound name | RI | peak l/min | peak l/s | library formula | library probability | CAS No. | Mol. Wt. | I (%) | II (%) | III (%) | IV (%) | V (%) |
|---|---|---|---|---|---|---|---|---|---|---|---|---|---|
| | sesquiterpenes | | | | | | | | | | | | |
| 1 | aromadendrene | 1533 | 12.72 | 3.27 | $C_{15}H_{24}$ | 82 | 489394 | 204 | 0.0015 ± 0.0010 | not detected | not detected | 0.6923 ± 0.5293 | not detected |
| 2 | guaiol | 1663 | 15.28 | 5.16 | $C_{15}H_{26}O$ | 79 | 489861 | 222 | 5.3112 ± 3.7519 | not detected | not detected | 2.3544 ± 0.1645 | 4.2907 ± 3.8226 |
| 3 | longifolenaldehyde | 1497 | 12.48 | 3.55 | $C_{15}H_{24}O$ | 73 | 19890847 | 220 | not detected | not detected | not detected | 1.3292 ± 0.9446 | not detected |
| 4 | espatulenol | 1497 | 12.48 | 2.52 | $C_{15}H_{24}O$ | 76 | 6750603 | 220 | 1.7892 ± 0.1246 | 2.3159 ± 1.7728 | not detected | 1.1377 ± 0.9486 | 1.1186 ± 0.6457 |
| 5 | 8-(hydroxymethyl)-3,6,8-trimethyloctahydro-1H-3a,7-methanoazulen-6-ol | 1975 | 20.77 | 0.37 | $C_{15}H_{26}O_2$ | 73 | 62600059 | 238 | not detected | not detected | 0.7556 ± 0.6302 | 0.0354 ± 0.0190 | 0.0409 ± 0.0230 |
| 6 | 1,1,4a,7-tetramethyl-2,3,4,4a,5,6,7,8-octahydro-1H-benzo[7]annulen-7-ol | 2140 | 26.13 | 2.57 | $C_{15}H_{26}O$ | 68 | 6892804 | 222 | not detected | 0.0759 ± 0.0203 | not detected | not detected | not detected |
| 7 | α-eudesmol | 1634 | 14.82 | 4.24 | $C_{15}H_{26}O$ | 83 | 1209718 | 222 | 0.7001 ± 0.2812 | 0.1698 ± 0.0071 | 0.9775 ± 0.1515 | 1.0196 ± 0.0624 | 0.7444 ± 0.6523 |
| 8 | 2,4a,5,8a-tetramethyl-1,2,3,4,4a,7,8,8a-octahydronaphthalen-1-yl acetate | 1712 | 16.1 | 5.27 | $C_{16}H_{26}O_2$ | 69 | 20558218 | 250 | 5.5941 ± 1.6811 | not detected | not detected | 6.8546 ± 3.4922 | 11.0327 ± 9.7659 |
| 9 | 5,6-dimethyl-8-isopropenylbicyclo(4.4.0)dec-1-en-3-one | 1744 | 16.68 | 3.77 | $C_{15}H_{22}O$ | 69 | 4674504 | 218 | not detected | 3.2207 ± 2.8745 | not detected | not detected | not detected |
| 10 | (+)-valencene | 1511 | 12.72 | 2.45 | $C_{15}H_{24}$ | 81 | 4630073 | 204 | not detected | 0.1616 ± 0.0175 | not detected | not detected | not detected |
| 11 | (3,8,8-trimethyl-1,2,3,4,5,6,7,8-octahydro-2-naphthalenyl)methyl acetate | 1719 | 16.22 | 0.61 | $C_{16}H_{26}O_2$ | 68 | 314773278 | 250 | not detected | not detected | 3.6388 ± 0.0790 | not detected | 7.3226 ± 0.0343 |
| 12 | dehydrofukinone | 1820 | 18.08 | 2.38 | $C_{15}H_{22}O$ | 78 | 19598459 | 218 | not detected | 3.7005 ± 2.7568 | not detected | 0.6524 ± 0.1635 | not detected |
| 13 | α-copaen-11-ol | 1685 | 15.63 | 5.89 | $C_{15}H_{24}O$ | 73 | 41370563 | 220 | 5.2734 ± 1.7874 | 7.1112 ± 5.6153 | 4.0888 ± 1.5279 | 1.9115 ± 1.4793 | 0.0369 ± 0.0147 |
| 14 | agaruspirol | 1663 | 15.28 | 4.49 | $C_{15}H_{26}O$ | 83 | 1460737 | 222 | not detected | 4.1321 ± 1.6386 | not detected | 6.2798 ± 3.1912 | not detected |
| 15 | 2-isopropenyl-4,4,6b-trimethyl-4,5,5a,6,6a,6b-hexahydro-2H-cyclopropa[g][1]benzofuran | 1646 | 15.87 | 4.84 | $C_{15}H_{22}O$ | 76 | 102681492 | 218 | not detected | 28.1937 ± 4.0029 | not detected | 3.6005 ± 0.2307 | 1.3055 ± 1.0888 |
| 16 | dehydrosaussurea lactone | 2102 | 24.73 | 4.71 | $C_{15}H_{20}O_2$ | 74 | 28290359 | 232 | not detected | 1.2429 ± 0.0490 | not detected | 0.0406 ± 0.0223 | 0.1199 ± 0.0372 |
| 17 | methyl-2-(3,6-dimethyl-2-oxo-6-vinyl-2,4,5,6,7,7a-hexahydrobenzofuran-5-yl)acrylate | 1867 | 18.9 | 3.02 | $C_{16}H_{20}O_4$ | 61 | 19892194 | 276 | not detected | not detected | not detected | 0.2207 ± 0.1593 | not detected |

**Table 1.** (*Continued.*)

| no. | compound name | RI | peak I/min | peak II/s | library formula | library probability | CAS No. | Mol. Wt. | I (%) | II (%) | III (%) | IV (%) | V (%) |
|---|---|---|---|---|---|---|---|---|---|---|---|---|---|
| 18 | 2,2,4-trimethyl-4-(2-methylallyl)hexahydrocyclo propa[cd]pentalene-1,3-dione | 1854 | 18.67 | 2.21 | $C_{15}H_{20}O_2$ | 74 | 94609184 | 232 | 0.1986 ± 0.1178 | not detected | 0.7312 ± 0.1133 | 0.0796 ± 0.0548 | 0.1786 ± 0.1174 |
| 19 | 3,4,4-trimethyl-3-[(1E)-3-methyl-1,3-butadienyl]bicyclo[4.1.0]heptan-2-one | 1867 | 18.9 | 2.72 | $C_{15}H_{22}O$ | 69 | 102146816 | 218 | not detected | not detected | not detected | 0.1445 ± 0.0096 | not detected |
| 20 | (E)-2,2,6-trimethyl-1-(4-methylpenta-2,4-dien-2-yl)-7-oxabicyclo[4.1.0]heptane | 2028 | 22.17 | 4.51 | $C_{15}H_{24}O$ | 68 | 89128143 | 220 | not detected | not detected | not detected | 0.1581 ± 0.0127 | 0.5011 ± 0.1012 |
| 21 | 1-[(1E)-2,3-Dimethyl-1,3-butadienyl]-2,2,6-trimethyl-7-oxabicyclo[4.1.0]heptane | 2028 | 22.17 | 5.76 | $C_{15}H_{24}O$ | 66 | 59744126 | 220 | not detected | not detected | not detected | 1.5545 ± 0.8055 | not detected |
| 22 | 2,6-dimethyl-6-(4-methyl-3-pentenyl)-2-cyclohexene-1-carbaldehyde | 1887 | 19.25 | 1.13 | $C_{15}H_{24}O$ | 72 | 56772077 | 220 | not detected | 0.9927 ± 0.7353 | not detected | not detected | not detected |
| 23 | (E)-2,3,3-trimethyl-2-(3-methylbuta-1,3-dien-1-yl)-6-methylenecyclohexan-1-one | 2208 | 28.58 | 2.65 | $C_{15}H_{22}O$ | 71 | 77822572 | 218 | not detected | not detected | not detected | not detected | 1.2203 ± 0.7010 |
| 24 | (E)-2,2,6-trimethyl-1-(3-methylbuta-1,3-dien-1-yl)-5-methylene-7-oxabicyclo[4.1.0]heptane | 2297 | 31.62 | 2.06 | $C_{15}H_{22}O$ | 70 | 70038209 | 218 | not detected | not detected | not detected | not detected | 1.2714 ± 0.9411 |
| 25 | 2-(3-isopropenyl-4-methyl-4-vinylcyclohexyl)-2-propanol | 1678 | 15.52 | 1.61 | $C_{15}H_{26}O$ | 68 | 639996 | 222 | not detected | not detected | not detected | not detected | 0.4491 ± 0.3673 |
| | total sesquiterpenes | | | | | | | | 18.8681 | 51.2570 | 10.1919 | 28.0654 | 29.6327 |
| | *monoterpenes* | | | | | | | | | | | | |
| 26 | 1,3,3-trimethyl-2-vinylcyclohexene | 1706 | 15.98 | 5.27 | $C_{11}H_{18}$ | 73 | 5293903 | 150 | 6.7032 ± 4.6849 | not detected | 0.7544 ± 0.4255 | 1.8548 ± 0.0363 | not detected |
| 27 | (E)-b-ionone | 1840 | 18.43 | 6.25 | $C_{13}H_{20}O$ | 66 | 79776 | 192 | 4.8090 ± 4.0616 | not detected | not detected | not detected | 0.2716 ± 0.0292 |
| 28 | (4E)-4-(2,6,6-trimethyl-2-cyclohexen-1-ylidene)-2-butanone | 2147 | 26.37 | 2.38 | $C_{13}H_{20}O$ | 66 | 56052610 | 192 | not detected | not detected | not detected | not detected | 1.5186 ± 0.8964 |
| 29 | 6,7-dimethyl-1,2,3,5,8,8a-hexahydronaphthalene | 1275 | 9.1 | 2.4 | $C_{12}H_{18}$ | 74 | 107914921 | 162 | not detected | 0.8786 ± 0.2955 | 2.0628 ± 0.8246 | not detected | not detected |
| 30 | 1,8-dimethyl-4,11-dioxatricyclo[6.2.1.02,7]undeca-2,9-diene | 1591 | 14.12 | 5.65 | $C_{11}H_{14}O_2$ | 70 | 121029638 | 178 | not detected | not detected | 1.6647 ± 0.2766 | not detected | not detected |
| 31 | (4-tert-butylphenyl)acetaldehyde | 1424 | 11.32 | 4.13 | $C_{12}H_{16}O$ | 73 | 109347457 | 176 | not detected | not detected | not detected | not detected | 0.0275 ± 0.0068 |
| 32 | thymol | 1762 | 17.03 | 4.09 | $C_{10}H_{14}O$ | 70 | 89838 | 150 | not detected | 8.3843 ± 7.7453 | not detected | not detected | not detected |

(*Continued.*)

**Table 1.** (*Continued.*)

| no. | compound name | RI | peak l/min | peak l/s | library formula | library probability | CAS No. | Mol. Wt. | I (%) | II (%) | III (%) | IV (%) | V (%) |
|---|---|---|---|---|---|---|---|---|---|---|---|---|---|
| | total monoterpenes | | | | | | | | 11.512 | 9.2629 | 4.4819 | 1.8548 | 1.8177 |
| | *diterpenes* | | | | | | | | | | | | |
| 33 | 2-methylenecholestan-3-ol | 1854 | 18.67 | 3.33 | $C_{28}H_{48}O$ | 70 | 2559968 | 400 | 0.0810 ± 0.0347 | 0.0094 ± 0.030 | not detected | not detected | 0.0674 ± 0.0061 |
| 34 | stigmast-5-en-3-yl acetate | 1752 | 17.73 | 1.66 | $C_{31}H_{52}O_2$ | 71 | 915059 | 456 | not detected | not detected | not detected | not detected | 1.1457 ± 0.4734 |
| | total diterpenes | | | | | | | | 0.0810 | 0.0094 | 0.0000 | 0.0000 | 1.2131 |
| | *aromatics* | | | | | | | | | | | | |
| 35 | 4,6-di-tert-butyl-o-cresol | 1490 | 12.37 | 1.67 | $C_{15}H_{24}O$ | 68 | 616557 | 220 | 0.1980 ± 0.0642 | not detected | 0.6796 ± 0.1222 | 0.5806 ± 0.0178 | 0.7009 ± 0.1337 |
| 36 | p-methoxybenzylacetone | 1524 | 12.95 | 3.5 | $C_{11}H_{14}O_2$ | 80 | 104201 | 178 | not detected | 0.4269 ± 0.0940 | not detected | 0.1421 ± 0.0612 | 0.4741 ± 0.2739 |
| 37 | (1-methoxy-4-methyl-3-pentenyl)benzene | 1921 | 19.83 | 1.91 | $C_{13}H_{18}O$ | 72 | 68705862 | 190 | not detected | 0.2839 ± 0.0257 | 2.4052 ± 1.6184 | 3.3340 ± 0.4246 | 1.4515 ± 0.3223 |
| 38 | (2,6,6-trimethylcyclohex-1-enylmethanesulfonyl) benzene | 2048 | 22.87 | 2.51 | $C_{16}H_{22}O_3S$ | 68 | 56691748 | 278 | not detected | not detected | not detected | 0.2523 ± 0.0559 | 0.7627 ± 0.2472 |
| 39 | 4-phenylbutan-2-one | 1267 | 8.98 | 2.3 | $C_{10}H_{12}O$ | 86 | 2550267 | 148 | not detected | 0.4726 ± 0.1153 | not detected | 3.5987 ± 2.6857 | not detected |
| 40 | dibutyl phthalate | 1961 | 20.53 | 5.83 | $C_{16}H_{22}O_4$ | 74 | 84742 | 278 | 7.3164 ± 4.8192 | not detected | not detected | not detected | not detected |
| 41 | ethyl benzoate | 1195 | 7.93 | 2.01 | $C_9H_{10}O_2$ | 85 | 93890 | 150 | not detected | 0.5335 ± 0.1159 | not detected | not detected | not detected |
| 42 | benzothiazole | 1259 | 8.87 | 2.79 | $C_7H_5NS$ | 83 | 95169 | 135 | not detected | 0.2800 ± 0.0481 | not detected | not detected | not detected |
| 43 | 6-(benzyloxy)-4,4-dimethylchroman-2-one | 1627 | 14.7 | 2.28 | $C_{18}H_{18}O_3$ | 72 | 84945108 | 282 | not detected | 0.2019 ± 0.0437 | not detected | not detected | not detected |
| | total aromatics | | | | | | | | 7.5144 | 2.1988 | 3.0848 | 7.9077 | 3.3892 |
| | *alkanes* | | | | | | | | | | | | |
| 44 | 4,7-dimethylundecane | 1065 | 6.07 | 1.2 | $C_{13}H_{28}$ | 85 | 17301325 | 184 | not detected | 0.7911 ± 0.3807 | 7.8485 ± 5.9292 | 2.3577 ± 0.3961 | 1.1704 ± 0.253 |
| 45 | tridecane | 1227 | 8.4 | 1.28 | $C_{13}H_{28}$ | 87 | 629505 | 184 | not detected | not detected | not detected | 1.4251 ± 0.2479 | not detected |
| 46 | hexadecane | 1299 | 9.45 | 2.53 | $C_{16}H_{34}$ | 84 | 544763 | 226 | 7.5238 ± 5.2513 | not detected | 5.7390 ± 4.5180 | 1.5395 ± 0.0409 | 2.3853 ± 2.0450 |
| 47 | 3,8-dimethylundecane | 1490 | 12.37 | 1.99 | $C_{13}H_{28}$ | 80 | 17301303 | 184 | 2.9744 ± 0.4509 | 0.6414 ± 0.2790 | 1.6418 ± 0.4401 | 0.6497 ± 0.3586 | 2.0775 ± 0.2829 |
| 48 | 2,6,10,15-tetramethylheptadecane | 1497 | 12.48 | 3.21 | $C_{21}H_{44}$ | 80 | 54833486 | 296 | not detected | not detected | not detected | 0.2712 ± 0.0094 | not detected |
| 49 | 3-ethyl-3-methylheptane | 1557 | 13.53 | 3.8 | $C_{10}H_{22}$ | 78 | 17302011 | 142 | not detected | not detected | not detected | 0.3293 ± 0.1140 | not detected |
| 50 | heptatriacontan-1-ol | 1935 | 20.07 | 3.38 | $C_{37}H_{76}O$ | 75 | 105794589 | 536 | 0.2469 ± 0.0489 | 0.0218 ± 0.0093 | 2.5082 ± 0.0568 | 0.0594 ± 0.0109 | 0.8715 ± 0.7472 |
| 51 | undecane | 1212 | 8.17 | 1.34 | $C_{11}H_{24}$ | 86 | 1120214 | 156 | 8.3629 ± 1.7491 | not detected | not detected | not detected | not detected |

(*Continued.*)

**Table 1.** (*Continued.*)

| no. | compound name | RI | peak l/min | peak ll/s | library formula | library probability | CAS No. | Mol. Wt. | I (%) | II (%) | III (%) | IV (%) | V (%) |
|---|---|---|---|---|---|---|---|---|---|---|---|---|---|
| 52 | 2,5,9-trimethyldecane | 1227 | 8.4 | 1.35 | $C_{13}H_{28}$ | 73 | 62108229 | 184 | not detected | not detected | not detected | not detected | 0.1544 ± 0.0072 |
|  | total alkanes |  |  |  |  |  |  |  | 19.1080 | 1.4543 | 17.7375 | 6.6319 | 6.6591 |
|  | *others* |  |  |  |  |  |  |  |  |  |  |  |  |
| 53 | 2-(7-Heptadecynyloxy)tetrahydro-2H-pyran | 1814 | 17.97 | 5.88 | $C_{22}H_{40}O_2$ | 69 | 56599509 | 336 | 0.2877 ± 0.0293 | 0.0747 ± 0.0176 | 0.2114 ± 0.1161 | 0.2749 ± 0.0608 | 0.3608 ± 0.1783 |
| 54 | 1-chlorooctadecane | 1935 | 20.07 | 3.34 | $C_{18}H_{37}Cl$ | 72 | 3386332 | 288 | not detected | not detected | not detected | 0.2630 ± 0.0122 | 0.0911 ± 0.0270 |
| 55 | 2,4-decadienal | 1331 | 9.92 | 2.89 | $C_{10}H_{16}O$ | 84 | 2363884 | 152 | not detected | not detected | 0.1276 ± 0.0761 | not detected | 0.1524 ± 0.0520 |
| 56 | methyl 2,5-octadecadiynoate | 2134 | 25.9 | 5.23 | $C_{19}H_{30}O_2$ | 72 | 57156919 | 290 | not detected | not detected | 0.4789 ± 0.0620 | not detected | 0.0159 ± 0.0075 |
| 57 | bicyclohexyliden-2-one | 2156 | 26.72 | 2.7 | $C_{12}H_{18}O$ | 68 | 1011127 | 178 | not detected | not detected | not detected | not detected | 0.2567 ± 0.1567 |
| 58 | methyl arachidonate | 2225 | 29.17 | 4.19 | $C_{21}H_{34}O_2$ | 75 | 2566894 | 318 | not detected | not detected | not detected | not detected | 0.0108 ± 0.0025 |
| 59 | vinyl stearate | 1048 | 6.53 | 1.11 | $C_{20}H_{38}O_2$ | 59 | 111637 | 310 | not detected | 0.0900 ± 0.0214 | not detected | not detected | not detected |
| 60 | 6-(3,5-dimethylfuran-2-yl)-6-methylheptan-2-one | 1941 | 20.18 | 5.81 | $C_{14}H_{22}O_2$ | 65 | 90165096 | 222 | not detected | 0.9870 ± 0.3374 | not detected | not detected | not detected |

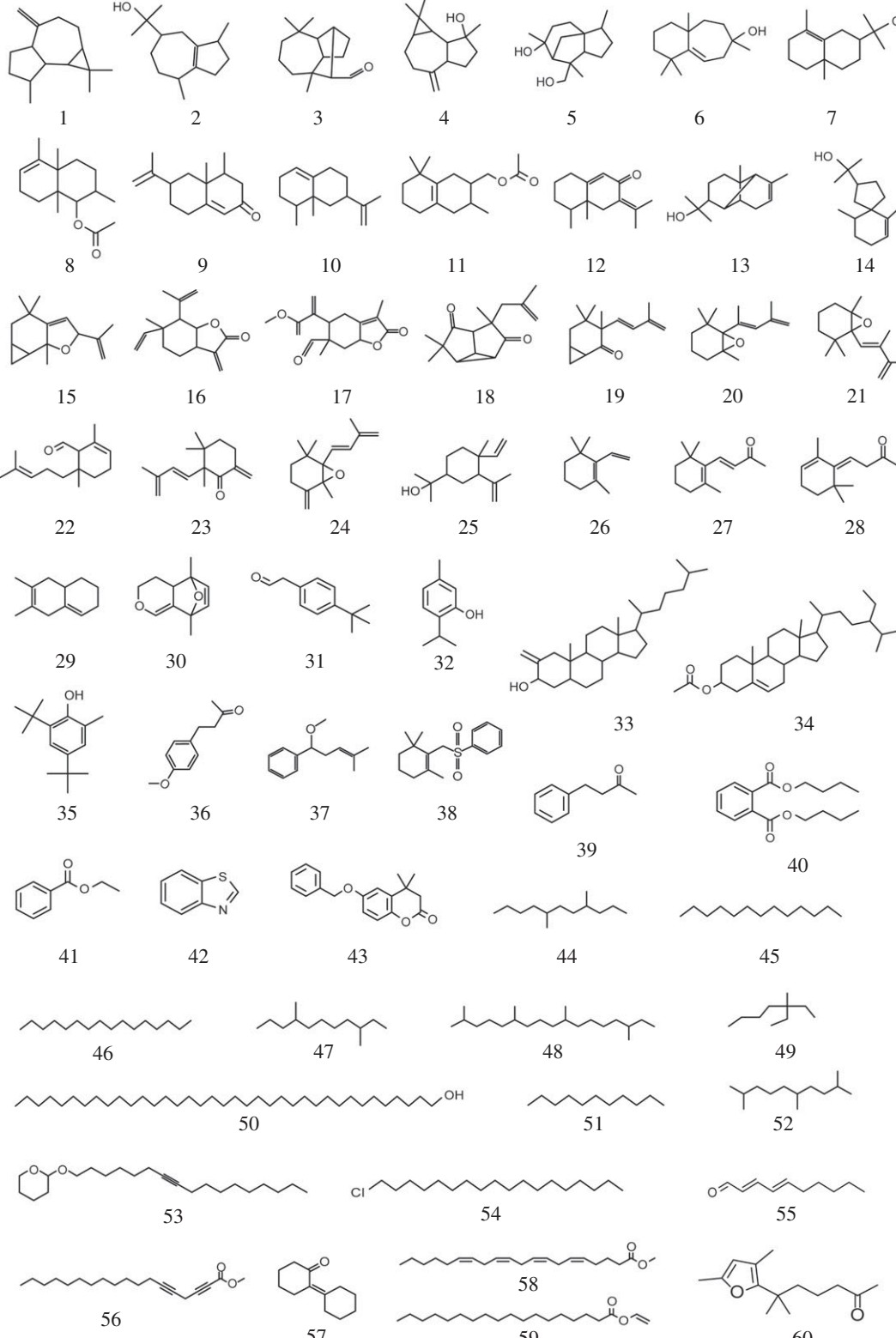

**Figure 2.** Structures of volatile compounds identified in the agarwood samples using GC × GC-HR-TOF-MS. Compound numbers follow the nomenclature used in table 1.

Type I than in the other types of agarwood (figure 7a). Among the 25 detected fungal classes, the Agaricomycetes, Archaeorhizomycets, Orbiliomycetes and Monoblepharidomycetes were most abundant in Type I, whereas the Tremellomycetes was most abundant in Type III, the Pezizomycetes and Taphrinomycetes were most abundant in Type II, and the Saccharomycetes, Dacrymycetes, Atractiellomycetes, Chytridiomycetes and Cystobasidiomycetes were most abundant in Type V.

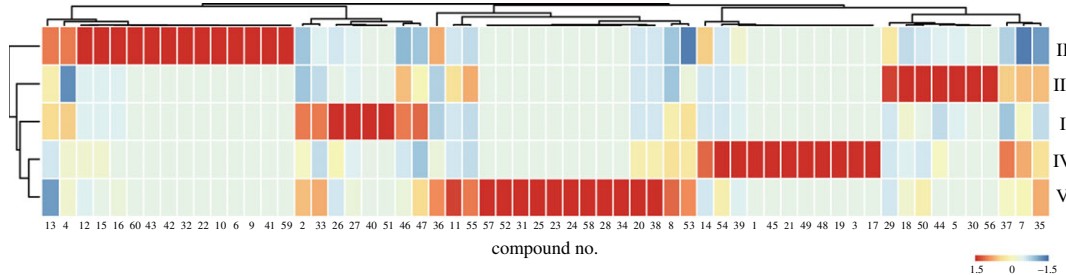

**Figure 3.** Distribution of identified compounds in the five types of agarwood.

**Table 2.** Diversity of fungal species in the five types of agarwood.

| type | no. | phylum | class | order | family | genus | OTUs |
|------|-----|--------|-------|-------|--------|-------|------|
| I | 1 | 5 | 20 | 41 | 71 | 64 | 378 |
|   | 2 | 5 | 19 | 43 | 77 | 67 | 350 |
|   | 3 | 4 | 16 | 40 | 68 | 61 | 359 |
| II | 1 | 4 | 18 | 43 | 73 | 66 | 337 |
|   | 2 | 3 | 15 | 39 | 63 | 56 | 251 |
|   | 3 | 4 | 17 | 37 | 66 | 57 | 285 |
| III | 1 | 2 | 12 | 31 | 47 | 44 | 298 |
|   | 2 | 3 | 15 | 35 | 55 | 54 | 352 |
|   | 3 | 4 | 15 | 39 | 62 | 60 | 339 |
| IV | 1 | 3 | 14 | 33 | 48 | 43 | 235 |
|   | 2 | 3 | 16 | 35 | 50 | 41 | 229 |
|   | 3 | 2 | 12 | 31 | 49 | 40 | 221 |
| V | 1 | 3 | 14 | 33 | 61 | 54 | 239 |
|   | 2 | 3 | 18 | 40 | 68 | 59 | 272 |
|   | 3 | 5 | 18 | 41 | 62 | 59 | 319 |

By contrast, no fungal classes were particularly abundant in Type IV (figure 7*b*). Among the 10 most abundant fungal classes, the Sordariomycetes was dominant in Types I, III and IV, whereas the Dothideomycetes and Eurotiomycetes were dominant in Types II and V, respectively. The fungal orders and families of Type I were different to those of Types II, III, IV and V (figure 7*c,d*), and the dominant fungal orders and families of the five agarwood types were different (figure 7*b,c*). The most dominant genera of Types V, I, III and II/IV were *Lasiodiplodia*, *Hydnellum*, *Phaeoisaria* and *Ophiocordyceps*, respectively (figure 7*e*).

## 3.5. Correlation between agarwood chemistry and fungal diversity

The correlation between the chemical and fungal components of the agarwood was analysed to further examine the correlation between fungal species and agarwood composition. Based on the presence of different volatiles and fungal genera, correlation analysis revealed that the Zygomycota was associated with the distribution of aromatic compounds ($r = 0.661^{**}$, $p < 0.01$; table 3). Correlations between the 30 most abundant fungal genera and levels of 60 compounds were analysed (table 4 and electronic supplementary material, table S1). Analysis of the 25 sesquiterpenes revealed that the presence of *Paraconiothyrium* and *Cladosporium* species was significantly correlated with compounds 11, 20 and 24; whereas the presence of *Limacella* and *Trichothecium* species was associated with compounds 24 and 25; the presence of *Resinicium* species was associated with compounds 23–25; and the presence of *Lasiodiplodia* species was associated with compound 23 ($r > 0.8$, $p < 0.01$; table 4 and table S1). Meanwhile, further investigation of the seven identified monoterpenes revealed that the presence of *Cercopemyces* and *Neofabraea* species was associated with compound 26; the presence of *Hydnellum*,

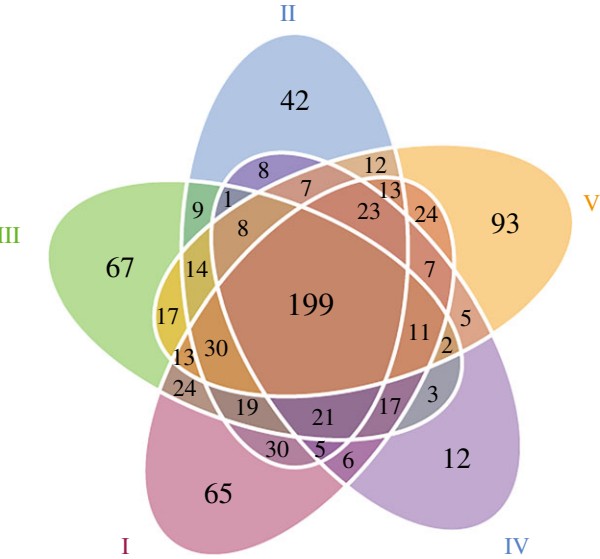

**Figure 4.** Distribution of fungal species among the five types of agarwood. Each circle represents one type of agarwood, and overlapping regions indicate common operational taxonomic units (OTUs); non-overlapping regions indicate type-specific OTUs.

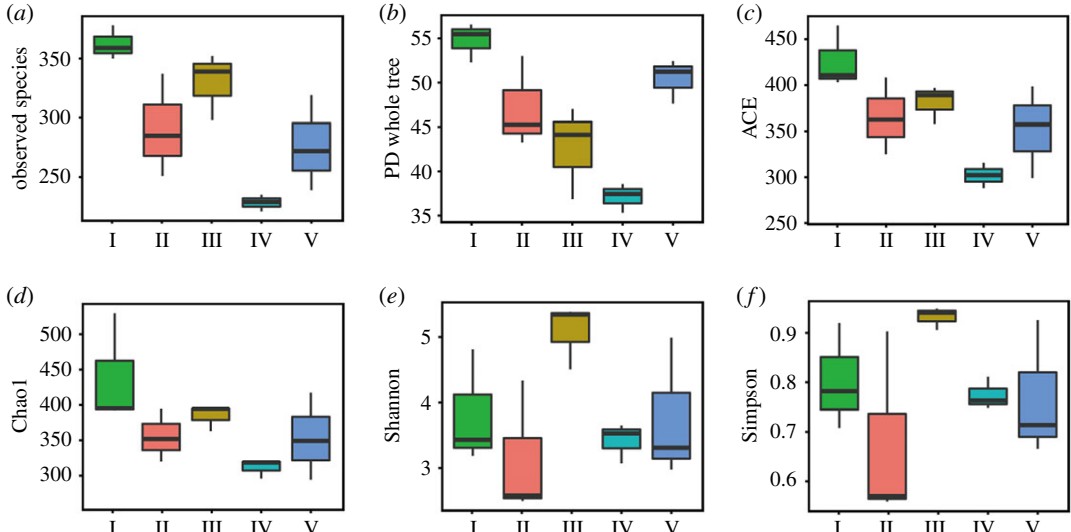

**Figure 5.** Richness and diversity of fungal communities associated with the five types of agarwood. The indices included observed species (*a*), PD whole tree (*b*), ACE (*c*), Chao 1 (*d*), Shannon (*e*) and Simpson (*f*) indices.

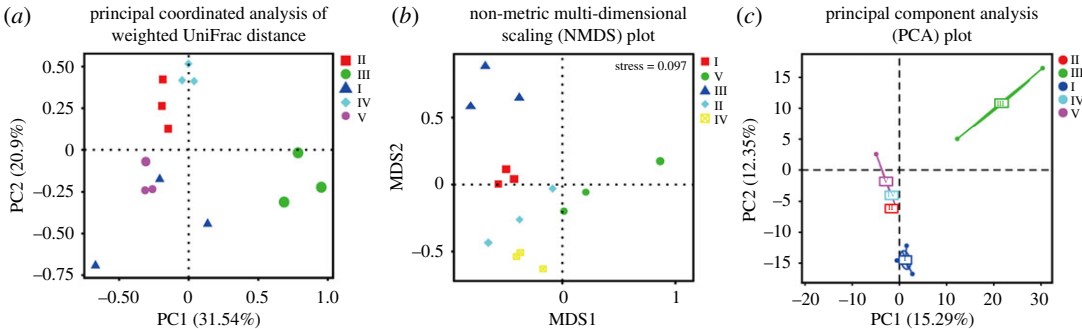

**Figure 6.** Evaluation of fungal communities by UniFrac analysis: (*a*) principal coordinated analysis of weighted UniFrac distance, which depicts the species composition structure of the five types of agarwood; (*b*) non-metric multi-dimensional scaling (NMDS) plot, which depicts the diversity of the fungal community by nonlinear structure; (*c*) principal component analysis (PCA) of the five types of agarwood.

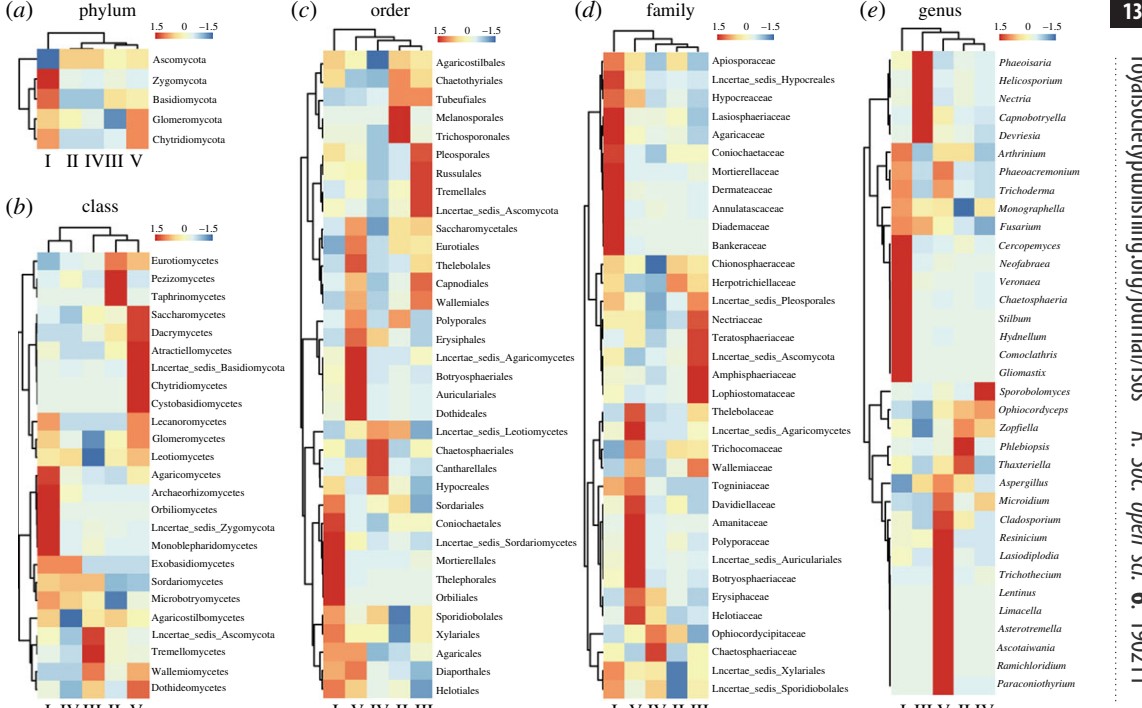

**Figure 7.** Distribution of fungal taxa in the five types of agarwood. Heat maps are based on the distribution of fungal phyla (*a*), classes (*b*), orders (*c*), families (*d*) and genera (*e*).

*Monographella* and *Veronaea* species was correlated with compound 27; the presence of *Limacella*, *Lasiodiplodia*, *Ascotaiwania*, *Asterotremella* and *Ramichloridium* species was correlated with compound 28; the presence of *Helicosporium* and *Phlebiopsis* species was associated with compounds 29 and 32, respectively; the presence of both *Phaeoisaria* and *Capnobotryella* species was correlated with compound 30; and the presence of *Lentinus* and *Resinicium* species was associated with compound 31 ($r > 0.8$, $p < 0.01$; table 4 and table S1). In regard to diterpene distribution, six fungal genera were found to be correlated with compound 34, including *Limacella*, *Lasiodiplodia*, *Lentinus*, *Resinicium*, *Trichothecium* and *Ramichloridium*, whereas *Phaeoacremonium* was associated with compound 33 ($r > 0.8$, $p < 0.01$; table 4 and table S1). Of the nine aromatics, compounds 38, 39, 40, and 42 were correlated with the presence of *Microidium*, *Sporobolomyces*, *Gliomastix* and *Veronaea*, and *Thaxteriella*, respectively ($r > 0.8$, $p < 0.01$; table 4 and table S1). Among the nine alkanes, compound 52 was associated with the presence of *Paraconiothyrium*, *Ascotaiwannia* and *Phaeoacremonium*, whereas compounds 44, 49 and 51 were associated with *Devriesin*, *Sporobolomyces* and *Comoclathris*, respectively ($r > 0.8$, $p < 0.01$; table 4 and table S1).

Together, these analyses indicate that the presence of *Lasiodiplodia*, *Cladosporium*, *Resinicium*, *Hydnellum* and *Monographella* is highly correlated with volatiles in eight common fungal genera that were present in all agarwood samples. Although *Fusarium*, *Trichoderma* and *Aspergillus* were not highly correlated with the volatiles ($r < 0.8$), the genera have been reported to play roles in agarwood formation [42–45].

# 4. Discussion

Agarwood is a typical example of the raw materials for perfume and oriental medicine, with the most potent compounds being sesquiterpene, monoterpene and aromatic compounds [3]. Natural agarwood can be divided into different types, owing to the different formation locations and incenses. The present study is the first work to investigate the chemical characteristics of agarwood types formed at different wound locations. Using GC × GC-HR-TOF-MS, the present study determined that differences in the type and number of compounds in the different types of agarwood were significant and that GC × GC-HR-TOF-MS can be used as an effective tool for the separation and identification of individual compounds from complex volatile oils.

**Table 3.** Correlation of volatiles and fungal phyla among the five types of agarwood.

| | sesquiterpenes | monoterpenes | diterpenes | aromatics | alkanes | Zygomycota | Chytridiomycota | Ascomycota | Glomeromycota | Basidiomycota |
|---|---|---|---|---|---|---|---|---|---|---|
| Sesquiterpenes | 1.000 | 0.161** | 0.099** | −0.010 | −0.727** | −0.167** | −0.059** | 0.289** | 0.324** | −0.288** |
| monoterpenes | 0.161** | 1.000 | −0.326** | −0.176** | 0.112** | 0.191** | −0.058** | −0.335** | −0.058** | 0.334** |
| diterpenes | 0.099** | −0.326** | 1.000 | −0.193** | −0.199** | −0.137** | 0.354** | 0.070** | 0.310** | −0.070** |
| aromatics | −0.010 | −0.176** | −0.193** | 1.000 | −0.016* | 0.661** | −0.149** | −0.288** | 0.296** | 0.279** |
| alkanes | −0.727** | 0.112** | −0.199** | −0.016* | 1.000 | 0.189** | 0.300** | −0.481** | −0.167** | 0.480** |
| Zygomycota | −0.167** | 0.191** | −0.137** | 0.661** | 0.189** | 1.000 | 0.094** | −0.286** | 0.145** | 0.272** |
| Chytridiomycota | −0.059** | −0.058** | 0.354** | −0.149** | 0.300** | 0.094** | 1.000 | −0.204** | 0.607** | 0.200** |
| Ascomycota | 0.289** | −0.335** | 0.070** | −0.288** | −0.481** | −0.286** | −0.204** | 1.000 | −0.330** | −1.000** |
| Glomeromycota | 0.324** | −0.058** | 0.310** | 0.296** | −0.167** | 0.145** | 0.607** | −0.330** | 1.000 | 0.327** |
| Basidiomycota | −0.288** | 0.334** | −0.070** | 0.279** | 0.480** | 0.272** | 0.200** | −1.000** | 0.327** | 1.000 |

**$p < 0.01$; *$p < 0.05$; italic labelled number, correlation coefficient $r$-value greater than 0.6.

**Table 4.** Correlation between volatile compounds and fungal genera.

| compound | fungal genus | r-value |
|---|---|---|
| sesquiterpenes | | |
| compound 5 | Helicosporium | 0.85** |
| | Nectria | 0.91** |
| compound 10 | Thaxteriella | 0.93** |
| compound 11 | Paraconiothyrium | 0.89** |
| | Cladosporium | 0.87** |
| compound 12 | Phlebiopsis | 0.89** |
| compound 16 | Phlebiopsis | 1.00** |
| compound 18 | Helicosporium | 0.94** |
| compound 20 | Paraconiothyrium | 0.95** |
| | Cladosporium | 0.93** |
| | Phaeoacremonium | 0.83** |
| compound 22 | Phlebiopsis | 0.91** |
| compound 23 | Lasiodiplodia | 0.98** |
| | Lentinus | 0.87** |
| | Resinicium | 0.89** |
| compound 24 | Paraconiothyrium | 0.85** |
| | Limacella | 0.81** |
| | Cladosporium | 0.85** |
| | Resinicium | 0.87** |
| | Trichothecium | 0.97** |

| compound | fungal genus | r-value |
|---|---|---|
| monoterpenes | | |
| compound 28 | Limacella | 0.81** |
| | Lasiodiplodia | 0.98** |
| | Ascotaiwania | 0.91* |
| | Asterotremella | 0.91** |
| | Ramichloridium | 0.90** |
| compound 29 | Helicosporium | 0.81** |
| compound 30 | Phaeoisaria | 0.86** |
| | Capnobotryella | 0.87** |
| compound 31 | Lentinus | 0.98** |
| | Resinicium | 0.92** |
| compound 32 | Phlebiopsis | 0.88** |
| diterpenes | | |
| compound 33 | Phaeoacremonium | 0.91** |
| compound 34 | Limacella | 0.93** |
| | Lasiodiplodia | 0.83** |
| | Lentinus | 0.85** |
| | Resinicium | 0.98** |
| | Trichothecium | 0.87** |
| | Tamichloridium | 0.83** |
| aromatics | | |
| compound 38 | Microidium | 0.94** |
| compound 39 | Sporobolomyces | 0.95** |

(Continued.)

**Table 4.** (*Continued.*)

| compound | | fungal genus | r-value | compound | | fungal genus | r-value |
|---|---|---|---|---|---|---|---|
| | compound 25 | Limacella | 0.99** | | compound 40 | Gliomastix | 0.91** |
| | | Ascotaiwania | 0.83** | | | Veronaea | 1.00** |
| | | Asterotremella | 0.83** | | compound 43 | Thaxteriella | 0.95** |
| | | Resinicium | 0.87** | alkanes | compound 44 | Devriesia | 0.86** |
| | | Trichothecium | 0.95** | | compound 49 | Sporobolomyces | 0.99** |
| | | Ramichloridium | 0.88** | | compound 51 | Comodathris | 0.93** |
| monoterpenes | compound 26 | Cercopemyces | 0.87** | | compound 52 | Paraconiothyrium | 0.99** |
| | | Neofabraea | 0.91** | | | Cladosporium | 0.98** |
| | compound 27 | Hydnellum | 0.85** | | | Phaeoacremonium | 0.89** |
| | | Veronaea | 0.92** | | | | |
| | | Monographella | 0.83** | | | | |

*p < 0.05; **p < 0.01.

Previous studies have reported that certain fungal strains obtained from *Aquilaria* can produce volatile compounds similar to those found in the essential oil of agarwood [46]. UniFrac analysis indicated that the crosscutting (including Types I and III) and rip cutting (Type IV) wounding methods affected the surrounding microbial communities, thereby indicating that regional wound differences may contribute to the variation observed in agarwood (figure 6). Distribution analysis revealed that the fungal communities of agarwood types vary significantly at sub-order levels of classification. Moreover, functional fungal genera were further investigated through heat map analysis of the 35 most abundant fungal genera (figure 7). Although the abundance of each fungal genus differed among the agarwood types, eight common genera were found in all the agarwood samples, including *Fusarium*, *Lasiodiplodia*, *Cladosporium*, *Trichoderma*, *Aspergillus*, *Resinicium*, *Hydnellum* and *Monographella*, which suggests that these eight fungal genera play key roles in agarwood formation. However, only *Lasiodiplodia* [21], *Fusarium* [47] and *Trichoderma* [48] fungi from *A. sinensis* have been shown to induce agarwood formation.

The results of the present study demonstrate that the chemical properties of *A. sinensis*-derived agarwood are strongly associated with fungal diversity. For example, *Lasiodiplodia* and *Cladosporium* were highly correlated with certain sesquiterpene, monoterpene and diterpene compounds (table 4 and table S1), whereas the abundances of *Hydnellum* and *Monographella* were correlated with certain monoterpene compounds. However, even though the presence of these five genera were highly correlated with the biosynthesis of agarwood volatiles, only *Lasiodiplodia* sp. has been reported that could promote agarwood formation. *Lasiodiplodia theobromae* isolated from *A. sinensis* has been reported to produce jasmonic acids that induce significant increases in sesquiterpene contents and chromate stimulation [49,50], which could, thereby, promote agarwood formation [21]. In the present study, the presence of *Fusarium*, *Trichoderma* and *Aspergillus* was also correlated with volatile contents (table S1), and other studies have also reported that the genera play roles in agarwood formation [42–45]. Indeed, *Fusarium* has been reported to induce the formation of agarwood in *A. malaccensis* and is capable of producing certain agarwood compounds, such as tridecanoic acid, α-santalol and spathulenol [43]. *Fusarium* has also been reported to produce specific secondary metabolites, such as pyrone derivatives [51]. Meanwhile, *Trichoderma* has been reported to promote the production of both sesquiterpenes and chromone derivatives in *A. malaccensis* cell suspension cultures [44], and *Aspergillus*, one of the most predominant fungal genera in agarwood, has been shown to induce the biosynthesis of mycotoxins, including aflatoxin B1 (AFB1) and ochratoxin A (OTA) [45]. Together, the results of the present study demonstrate that the chemical properties of *A. sinensis* are strongly associated with fungal diversity. However, it remains unclear whether the volatile constituents of agarwood are synthesized by the fungi or by the host trees. Accordingly, future research should focus more on the underlying mechanisms of agarwood formation, such as the role of functional genes in the interaction between *Aquilaria*, fungi and agarwood compounds.

# 5. Conclusion

The aim of the present study was to investigate the relationship between the chemistry and fungal associates of agarwood formed at different spatial locations. The findings presented here reveal that the location of agarwood formation significantly affects the chemical and fungal constituents of agarwood in *A. sinensis*. The occurrence of terpenoids, such as sesquiterpenes, monoterpenes and diterpenes, was closely related to fungal diversity, which is a primary determinant of agarwood properties. In agreement with previous studies that have reported that the volatile oil of agarwood can inhibit fungal growth, the results of the present study indicate that the volatile compounds of agarwood directly affect fungal diversity, which could further influence agarwood formation.

Data accessibility. Raw reads of the high-throughput sequencing data: Sequence Read Achieve (SRA) database (BioProject number: PRJNA509099).

Authors' contributions. L.H. and Y.Y. conceived and designed the study, and helped draft the manuscript. J.L. carried out the data analysis, participated in the design of the study and drafted the manuscript; X.Z. and J.Y. carried out the GC × GC-HR-TOF-MS experiment; J.Z. carried out the statistical analyses; C.J. and X.C. carried out the treatment of trees and selected the agarwood samples. All authors gave final approval for publication.

Competing interests. We declare we have no competing interests.

Funding. This work was supported by the National Natural Science Foundation of China (grant no. 81603236) and the Fundamental Research Funds for the Central Public Welfare Research Institutes (grant no. ZZXT201904).

Acknowledgements. The authors are grateful to Ou Huang, the general manager of Guangdong Shangzhengtang Group Co., for his assistance during *A. sinensis* planting, wound treatment, and agarwood sample collection.

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
