## [Reviewer comments · Royal Society Open Science]

Review History

RSOS-190211.R0 (Original submission)

Review form: Reviewer 1

Is the manuscript scientifically sound in its present form?

Yes

Are the interpretations and conclusions justified by the results?

No

Is the language acceptable?

No

Is it clear how to access all supporting data?

Yes

Do you have any ethical concerns with this paper?

No

Have you any concerns about statistical analyses in this paper?

No

Recommendation?

Major revision is needed (please make suggestions in comments)

Comments to the Author(s)

The study reported on the fungal diversity and volatile compounds present at different parts of the *Aquilaria sinensis* tree upon wounding treatment. The chemical compounds obtained were correlated with the fungal diversity based on different tree parts. The latter work could be considered as novel to agarwood-related studies. However, the manuscript was poorly presented and the authors should give more attention to their writings.

there are several important information the author left out when reporting their work; the authors need to enhance their presentation by referring to published papers for the description in MnM; the authors tend to put Discussion materials in their Results; the authors need to determine which Table and Figure are important to display in the manuscript and which are not. I suggest that this manuscript can be accepted if the below comments are attained.

Title

The authors only compared the fungal diversity and volatile compound between different tree parts, but the effects were not mentioned in the study.

Abstract

The abstract needs to be edited based on the comments below.

Introduction

The authors did not emphasize the importance of fungal species identification in *Aquilaria* tree. There is no information regarding endophytic and introduced fungi in man-made agarwood induction process. While the Fungi-*Aquilaria* relationship is a crucial key to agarwood formation, the authors should provide more information on the importance of fungi towards agarwood formation in *A. sinensis*. And I believe the finding for this study is to contribute assistance towards sustainable agarwood production in cultivated *A. sinensis*, hence such information should also be included.

MnM

Missing information on sampling techniques. Poor scientific write-up. The authors need to enhance their presentation by referring to published work for this section. I do not agree that the wounding treatment was carried out at different (spatial-formed) tissue types. The wounding points mentioned in the study were all from a single, or to breakdown, the most two tissue type: wood tissue from the stem and wood tissue from the branch. Scientifically, the five types of agarwood mentioned in this study are just names for the agarwood formed at different location on the tree stem. Instead, the authors should not mention the names of the agarwood in the MnM but to bring it up only at the Discussion section as an additional information on the naming practice among agarwood collectors in China. The samples, in my opinion, should be just treated as merely different wounding points.

Results

The authors tend to put Discussion materials in the Results section, which is incorrect.

Discussion

Poor discussed. With the amount of information obtained from the Results, the authors did not explain the possible causes of the strong correlation in certain fungi species and chemical compounds. There is possibility that the fungus is endophytic as well as introduced. furthermore, the samples were only collected at the end of the fifth year post-wounding treatment, this does not explain the correlation between the fungi and the agarwood formation because agarwood forms during the process. The process may have stopped at the 3rd or 4th year. My point is that to look at the true correlation between fungal diversity that would contribute towards agarwood formation, the authors should have carried out a time-lined sampling technique. Therefore, this study only explains the fungal diversity that is present when the agarwood has formed, and this does not directly provide information on the fungi whether they contribute towards the formation of agarwood in *A. sinensis*. This as well refers to the correlation between volatile compounds available and fungi diversity. In order to prove that the fungi are responsible to the great number of a certain volatile compound, there is a need to forward the study to specifically test on the efficacy of the fungus in contributing the production of a certain agarwood-related compound. Also, the authors should tone down their confidence between the volatile and fungi diversity recorded from their study as the study was only carried out in a single location. The fungal diversity could be affected by the environment, thus affect the result on the volatile compounds. Overall, the discussion needs to be refocused and rearranged to meet the standard of the study.

Conclusion

Figures & Tables

Removal and insertion of Figure and Tables are suggested in the comments below. The authors have to use good choice of language to present the Figure captions. If the data in the Figures or Tables were results from some software or database, there is a need to provide information of the said software or database. The Figures and Tables themselves should be able to be self-explanatory.

- >Line 45-49 sentence is too long
- >Line 59 emesia? or amnesia?
- >Line 51 and "is" mainly
- >Line 51 *Aquilaria sinensis* (Lour.) Spreng.
- >Line 51 reference 11 and 12 are studies on *Aquilaria crassna*, not *A. sinensis*
- >Line 54 endangered
- >Line 55 International
- >Line 56 "including *A. sinensis*" is redundant
- >Line 57 "continuous use" is inappropriate, suggest to change it as "sustainable production"
- >Line 58 "widely" cultivated
- >Line 59 "Due to" its rarity
- >Line 59-60 this is not a convincing statement. There are more than one species being rare and has crucial value that is widely cultivated, and why is only *A. sinensis* a good model?
- >Line 61, remove "usually"
- >Line 62 consequence of attack to what?
- >Line 62 promotes
- >Line 62 "Since" the last century
- >Line 64 Fungi species such as
- >Line 64 do not italic "sp."
- Line 69 reference 22 and 23 described the endophytic fungal diversity in different plant tissues (root, stem and leaf), while the study only involved the wood tissue in the stem and branch.

Furthermore the fungal diversity reported in this study could also involved those being introduced from the surrounding due to the wounding treatment. Thus, the references provided here do not reflect as a support to this study.

>Line 70 tissues of formation? incense? i do not understand what these mean.

>Line 70 citations?

>Line 71-76 i do not think this information should be in the Introduction. instead it is part of MnM. the authors should briefly give some information on agarwood formation on different tree parts published by previous studies, giving the readers a hint on their hypothesis of the relationship between fungal diversity and different wounded parts.

>Line 76-79 please rephrase the sentence.

>Line 80-81 chemical characteristics "found in agarwood formed in different wounding points inflicted on the tree".

>Line 82 system "for" agarwood

>Line 84 described

>Line 91 structures found in different parts of agarwood formation

>Line 92 When the authors mentioned to investigate the key fungi associated with different volatile compounds, I would be looking forward on a validation step where the fungi is isolated, then reintroduced back to the tree to confirm its strength to produce those volatile, yet there is no such experiment carried out in this study. hence, i think the authors need to rephrase this objective.

>Line 96-98 what is the sampling technique? wounded using what? when was this carried out?

how do you collect the formed agarwood? using what technique?

>Line 98-99 the number of individuals are unclear. each treatment has three individuals, so how many trees were there? how many treatments were there? is it 1 treatment 3 trees, then you got 5 different wounding points, so total is 15 tree? or u put all 5 wounding points on the same tree, then you have 3 biological replicates? Also, is there any negative control (a none wounded tree). How do you select your trees? are they side to side? or far apart? how trees interval? the information on the sampling technique is unclear. Please clarify.

>Line 99 the information in the Introduction should merge in here. However I have to raise my disagreement on the use of the name for the agarwood type as the specimen name, but to replace them by the organ/location of the tree where the wounding treatment is carried out as the specimen name. In such manner, the names for the type of agarwood form can be mentioned in the Discussion section. Unless the authors took the initiative to prepare an additional subsection in the MnM to specifically describe how the wounding treatment was carried out.

>Line 113 provide full name of ITS1, while its shortform in parenthesis. and i assume you amplify its sequence, not the whole gene.

>Line 114-115 provide complete sequences of the forward and reverse primers and their citation. and the reason why this primer set was chosen for this work was not mentioned.

>Line 113-123 This whole part has to be rewrite. The way it is written is not scientific. It is not mentioned that whose DNA was extracted and sequenced. please refer to published papers on how the methodology on the micro-array part should be presented. Machines should come with their brand name and country of make in parenthesis.

>Line 127 built? please provide name of software.

>Line 128 cut into what size?

>Line 130 remove "under filtering conditions according to", replace with "using".

>Line 132 compared "against" the

>Line 132 reference for Unite database.

>Line 133 "to" obtain

>Line 135 replace "by" with "using"

>Line 135 UPARSE, not Uparse.

>Line 136 reference for MOTHUR

>Line 137 replace "reached the" with "achieved"

>Line 137-138 QIIME is a program name, remove the full name in parenthesis and replace it with

its reference

- >Line 139 reference for SILVA, Greengene and RDP database
- >Line 139 replace "of" with "on"
- >Line 140 this normalized data output. What this? please be specific.
- >Line 141 replace "Diagrams of Venn analysis" with "Venn diagrams"
- >Line 141-142 some of the indexes and analyses come with their full name, please use their full names and put their shortnames in parenthesis
- >Line 147 put the weight of the sample in the sentence, not in parenthesis.
- >Line 147 2-mL
- >Line 148 remove "Then"
- >Line 150 reference for 40 kHz ultrasonic cold extraction method
- >Line 152 0.22-um
- >Line 153 stored in dark? or stored in a non-transparent?
- >Line 156-159 please check on the configuration, it is not properly written.
- >Line 159 took place "in"
- >Line 160 information about the column should be written in the sentence instead of in parenthesis. Only the brand name and country should be in parenthesis.
- >Line 163 under the following "setting"
- >Line 164-165 the use of symbol is inconsistent, should it be C min⁻¹ or C/min? also goes to Line 151.
- >Line 169 spectra per second? does it come with a symbol?
- >Line 170 expressed "in percentage of"...to "the" total
- >Line 172 authentic standards from where? any references? and what is Wiley? and please provide references and shortforms for these libraries.
- >Line 180 mentioned before, thus only shortform here
- >Line 182-183 using what program or technique?
- >Line 184 reference of software in parenthesis
- >Line 188 were first "identified using" the
- >Line 189-192 Figure 2A, 2B, Supplemental Figure S1 and S2 do not provide any useful information. suggest to remove.
- >Line 193 remove complete name for NIST as mentioned in Line 173
- >Line 193-194 60 compounds were tentatively identified with good match, which include Line 196 Supplemental Table S1 and Supplemental Figure S3 are useful and important information. They SHOULD NOT be supplemental data.
- >Line 196-197 this sentence should be in the MnM
- >Line 198 characterized by a high percentage of
- >Line 206 Figure 2C is irrelevant, it can be moved to the Supplementary data section.
- >Line 206-209 This should be in the Discussion section, not Results.
- >Line 211-212 This should be in the Discussion section, not Results.
- >Line 215 Figure 2D is irrelevant. The information can be presented in Supplementary Table S1
- >Line 218-220 This should be in the Discussion section, not Results.
- >Line 222 Figure 2E should be a standalone figure.
- >Line 222 I am not sure if this is correct, but how do you relate a heat map with a tree? should it be a group?
- >Line 223-227 This should not be in the Results.
- >Line 229-230 This should not be in the Results and replace "investigated" with "identified".
- >Line 231 how many agarwood samples? please state the number.
- >Line 233 Figure 3A should be replaced with a Table instead, with clear records of how many reads based on phylum, class, order, family and genus for all samples.
- >Line 234 with the Table proposed for Figure 3A, information in Figure 3B should be merged into the Table and Figure 3B would be irrelevant
- >Line 236 replaced "required" with "acquired"
- >Line 236 Figure 3C should be a standalone

>Line 238 Figure 4A should be break-downed into Figure 4A-F. description of x-axis in simpson is missing.

>Line 242-243 This sentence is not a Result, it is part of a Discussion

>Line 243-244 This is MnM

>Line 245-248 This is Discussion, while Results for Figure 4B and 4C were not described. Figure 4B and 4C should not go under Figure 4. They should both go under a different Figure together. Same goes to Supplementary Figure S4.

>Line 250-251 I personally think that the heat-map is sufficient to show the results for the taxonomical analysis. The bar-chart is not relevant. Thus Supplemental Figure S6 is important and should be included in the manuscript.

>Line 254 Basidiomycota

>Line 267-269 This is Discussion

>Line 275-279 This is Discussion

>Line 282-286 This is Discussion

>Line 292 aromatic compounds

>Line 292 I do not understand why your correlation plot contain both fungal and chemical compounds in the same axis. Are you planning to see the correlation between fungal species and chemical compounds present in the sample? While this is the novelty of your study, yet it is in Supplemental form. You should choose to display the Supplemental Table S2 in your manuscript, and replace the yellow boxes with and underline or bold fonts to show the cv.

>Line 294 Figure 6 is irrelevant, while Supplemental Table S3 is an important result. However the size of the Table makes it difficult to be presented in the manuscript. In my opinion, it should be included in the text and the decimal points should be reduce to 3 digits; the correlation cv should be marked with underline or bold fonts instead of yellow highlights; the fungi names should be arranged in alphabetical order; The Y-axis should be the fungi names; The X-axis should be the compound numbers (with a general name "compounds" on top and only numbers below to reduce the use of "compound" for every number).

>Line 323 This is Discussion

>Line 325 This is not Result, instead it looks like a conclusion in the Discussion.

>Line 338 I do not think there is a distinct spatial-formed tissue in this study. There are all wood tissues.

>Line 296 A Discussion sentence in a Conclusion is not appropriate.

Review form: Reviewer 2

Is the manuscript scientifically sound in its present form?

Yes

Are the interpretations and conclusions justified by the results?

Yes

Is the language acceptable?

Yes

Is it clear how to access all supporting data?

Yes

Do you have any ethical concerns with this paper?

No

Have you any concerns about statistical analyses in this paper?

No

Recommendation?

Major revision is needed (please make suggestions in comments)

Comments to the Author(s)

The present manuscript by Liu et al reports effects of fungal diversity on volatile compounds in different spatial formations of agarwood originated from *Aquilaria sinensis*. Authors have also showed relationship between fungal community and volatile compounds. The experiments are well conceived and well presented.

Major concerns:

- 1) where is reference of extraction method of agarwood oil? or Normally, the volatile oil is generally obtained from hrdrodistillation only. No discussion on extraction method relevant with extracting solvent provided in manuscript.
2. Use the common name for identified compounds. Kovat index is needed to present for identification confirmation. For example, mass spectrum of compounds 7 in Fig. 2 is not similar to those from library.
3. Fig S5B and Fig 5, other genus is the major group of fungi in all samples. The fungi in other genus may be correlated mainly to volatile compound production. The metagenome sequencing is important in this experiment. Genus of all isolated fungal strains are completely identified by PCR.
4. GCxGC chromatogram is not clear. Select one to present between 3D or 2D.

Decision letter (RSOS-190211.R0)

08-Apr-2019

Dear Dr Liu,

The editors assigned to your paper ("Effects of fungal diversity on volatile compounds in different spatial formations of agarwood originated from *Aquilaria sinensis*") have now received comments from reviewers. We would like you to revise your paper in accordance with the referee and Associate Editor suggestions which can be found below (not including confidential reports to the Editor). Please note this decision does not guarantee eventual acceptance.

Please submit a copy of your revised paper before 01-May-2019. Please note that the revision deadline will expire at 00.00am on this date. If we do not hear from you within this time then it will be assumed that the paper has been withdrawn. In exceptional circumstances, extensions may be possible if agreed with the Editorial Office in advance. We do not allow multiple rounds of revision so we urge you to make every effort to fully address all of the comments at this stage. If deemed necessary by the Editors, your manuscript will be sent back to one or more of the original reviewers for assessment. If the original reviewers are not available, we may invite new reviewers.

- Data accessibility

If you wish to submit your supporting data or code to Dryad (<http://datadryad.org/>), or modify your current submission to dryad, please use the following link:
<http://datadryad.org/submit?journalID=RSOS&manu=RSOS-190211>

- Competing interests

- Authors' contributions

- Acknowledgements

- Funding statement

Kind regards,

on behalf of Professor Xinguang Zhu (Associate Editor) and Professor Kevin Padian (Subject Editor)

Associate Editor's comments (Professor Xinguang Zhu):

Dear Dr. Huang,

The submission has been reviewed by two domain experts. Based on their recommendations and also my evaluation for the manuscript, I judge that this paper requires a major revision before it can be accepted for publication in RSOS. I feel that the novelty of the paper requires clarification. In addition, given that the observed correlation between the wood composition and also fungal diversity, authors suggest that this information can be used to guide future production of wood with altered wood quality. please describe in the text how this can be achieved in practice. In addition, the authors correlated a few wood types with the fungal species and then make the statement that the fungal diversity influence the wood type. It is equally possible that the fungal species diversity was generated by the difference wood composition. Which is the cause, which is the result? Please discuss this point. Please address all the other concerns of the reviewers as well.

Looking forward to receiving a revised manuscript.

Best wishes,

Xinguang Zhu

SIPPE

Associate Editor: 2

Comments to the Author:

I have read this paper and found the results very interesting. I have some comments on this study.

1. Please describe clearly the novelty of this paper.
2. Given that the observed correlation between the wood composition and also fungal diversity, authors suggest that this information can be used to guide future production of wood with altered wood quality. please describe in the text how this can be achieved in practice.
3. The authors correlated a few wood types with the fungal species and then make the statement that the fungal diversity influence the wood type. It is equally possible that the fungal species diversity was generated by the difference wood composition. Which is the cause, which is the result? Please discuss this point.

Comments to Author:

Reviewers' Comments to Author:

Reviewer: 1

Comments to the Author(s)

The study reported on the fungal diversity and volatile compounds present at different parts of the *Aquilaria sinensis* tree upon wounding treatment. The chemical compounds obtained were correlated with the fungal diversity based on different tree parts. The latter work could be considered as novel to agarwood-related studies. However, the manuscript was poorly presented and the authors should give more attention to their writings.

there are several important information the author left out when reporting their work; the authors need to enhance their presentation by referring to published papers for the description in MnM; the authors tend to put Discussion materials in their Results; the authors need to determine which Table and Figure are important to display in the manuscript and which are not. I suggest that this manuscript can be accepted if the below comments are attained.

Title

The authors only compared the fungal diversity and volatile compound between different tree parts, but the effects were not mentioned in the study.

Abstract

The abstract needs to be edited based on the comments below.

Introduction

The authors did not emphasize the importance of fungal species identification in *Aquilaria* tree. There is no information regarding endophytic and introduced fungi in man-made agarwood induction process. While the Fungi-*Aquilaria* relationship is a crucial key to agarwood formation, the authors should provide more information on the importance of fungi towards agarwood formation in *A. sinensis*. And I believe the finding for this study is to contribute assistance towards sustainable agarwood production in cultivated *A. sinensis*, hence such information should also be included.

MnM

Missing information on sampling techniques. Poor scientific write-up. The authors need to enhance their presentation by referring to published work for this section. I do not agree that the wounding treatment was carried out at different (spatial-formed) tissue types. The wounding points mentioned in the study were all from a single, or to breakdown, the most two tissue type: wood tissue from the stem and wood tissue from the branch. Scientifically, the five types of agarwood mentioned in this study are just names for the agarwood formed at different location on the tree stem. Instead, the authors should not mention the names of the agarwood in the MnM but to bring it up only at the Discussion section as an additional information on the naming practice among agarwood collectors in China. The samples, in my opinion, should be just treated as merely different wounding points.

Results

The authors tend to put Discussion materials in the Results section, which is incorrect.

Discussion

Poor discussed. With the amount of information obtained from the Results, the authors did not explain the possible causes of the strong correlation in certain fungi species and chemical compounds. There is possibility that the fungus is endophytic as well as introduced. furthermore,

the samples were only collected at the end of the fifth year post-wounding treatment, this does not explain the correlation between the fungi and the agarwood formation because agarwood forms during the process. The process may have stopped at the 3rd or 4th year. My point is that to look at the true correlation between fungal diversity that would contribute towards agarwood formation, the authors should have carried out a time-lined sampling technique. Therefore, this study only explains the fungal diversity that is present when the agarwood has formed, and this does not directly provide information on the fungi whether they contribute towards the formation of agarwood in *A. sinensis*. This as well refers to the correlation between volatile compounds available and fungi diversity. In order to prove that the fungi are responsible to the great number of a certain volatile compound, there is a need to forward the study to specifically test on the efficacy of the fungus in contributing the production of a certain agarwood-related compound. Also, the authors should tone down their confidence between the volatile and fungi diversity recorded from their study as the study was only carried out in a single location. The fungal diversity could be affected by the environment, thus affect the result on the volatile compounds. Overall, the discussion needs to be refocused and rearranged to meet the standard of the study.

Conclusion

Figures & Tables

Removal and insertion of Figure and Tables are suggested in the comments below. The authors have to use good choice of language to present the Figure captions. If the data in the Figures or Tables were results from some software or database, there is a need to provide information of the said software or database. The Figures and Tables themselves should be able to be self-explanatory.

>Line 45-49 sentence is too long

>Line 59 emesia? or amnesia?

>Line 51 and "is" mainly

>Line 51 *Aquilaria sinensis* (Lour.) Spreng.

>Line 51 reference 11 and 12 are studies on *Aquilaria crassna*, not *A. sinensis*

>Line 54 endangered

>Line 55 International

>Line 56 "including *A. sinensis*" is redundant

>Line 57 "continuous use" is inappropriate, suggest to change it as "sustainable production"

>Line 58 "widely" cultivated

>Line 59 "Due to" its rarity

>Line 59-60 this is not a convincing statement. There are more than one species being rare and has crucial value that is widely cultivated, and why is only *A. sinensis* a good model?

>Line 61, remove "usually"

>Line 62 consequence of attack to what?

>Line 62 promotes

>Line 62 "Since" the last century

>Line 64 Fungi species such as

>Line 64 do not italic "sp."

Line 69 reference 22 and 23 described the endophytic fungal diversity in different plant tissues (root, stem and leaf), while the study only involved the wood tissue in the stem and branch. Furthermore the fungal diversity reported in this study could also involved those being introduced from the surrounding due to the wounding treatment. Thus, the references provided here do not reflect as a support to this study.

>Line 70 tissues of formation? incense? i do not understand what these mean.

>Line 70 citations?

>Line 71-76 i do not think this information should be in the Introduction. instead it is part of MnM. the authors should briefly give some information on agarwood formation on different tree parts published by previous studies, giving the readers a hint on their hypothesis of the relationship between fungal diversity and different wounded parts.

>Line 76-79 please rephrase the sentence.

>Line 80-81 chemical characteristics "found in agarwood formed in different wounding points inflicted on the tree".

>Line 82 system "for" agarwood

>Line 84 described

>Line 91 structures found in different parts of agarwood formation

>Line 92 When the authors mentioned to investigate the key fungi associated with different volatile compounds, I would be looking forward on a validation step where the fungi is isolated, then reintroduced back to the tree to confirm its strength to produce those volatile, yet there is no such experiment carried out in this study. hence, i think the authors need to rephrase this objective.

>Line 96-98 what is the sampling technique? wounded using what? when was this carried out? how do you collect the formed agarwood? using what technique?

>Line 98-99 the number of individuals are unclear. each treatment has three individuals, so how many trees were there? how many treatments were there? is it 1 treatment 3 trees, then you got 5 different wounding points, so total is 15 tree? or u put all 5 wounding points on the same tree, then you have 3 biological replicates? Also, is there any negative control (a none wounded tree). How do you select your trees? are they side to side? or far apart? how trees interval? the information on the sampling technique is unclear. Please clarify.

>Line 99 the information in the Introduction should merge in here. However I have to raise my disagreement on the use of the name for the agarwood type as the specimen name, but to replace them by the organ/location of the tree where the wounding treatment is carried out as the specimen name. In such manner, the names for the type of agarwood form can be mentioned in the Discussion section. Unless the authors took the initiative to prepare an additional subsection in the MnM to specifically describe how the wounding treatment was carried out.

>Line 113 provide full name of ITS1, while its shortform in parenthesis. and i assume you amplify its sequence, not the whole gene.

>Line 114-115 provide complete sequences of the forward and reverse primers and their citation. and the reason why this primer set was chosen for this work was not mentioned.

>Line 113-123 This whole part has to be rewrite. The way it is written is not scientific. It is not mentioned that whose DNA was extracted and sequenced. please refer to published papers on how the methodology on the micro-array part should be presented. Machines should come with their brand name and country of make in parenthesis.

>Line 127 built? please provide name of software.

>Line 128 cut into what size?

>Line 130 remove "under filtering conditions according to", replace with "using".

>Line 132 compared "against" the

>Line 132 reference for Unite database.

>Line 133 "to" obtain

>Line 135 replace "by" with "using"

>Line 135 UPARSE, not Uparse.

>Line 136 reference for MOTHUR

>Line 137 replace "reached the" with "achieved"

>Line 137-138 QIIME is a program name, remove the full name in parenthesis and replace it with its reference

>Line 139 reference for SILVA, Greengene and RDP database

>Line 139 replace "of" with "on"

>Line 140 this normalized data output. What this? please be specific.

>Line 141 replace "Diagrams of Venn analysis" with "Venn diagrams"

- >Line 141-142 some of the indexes and analyses come with their full name, please use their full names and put their shortnames in parenthesis
- >Line 147 put the weight of the sample in the sentence, not in parenthesis.
- >Line 147 2-mL
- >Line 148 remove "Then"
- >Line 150 reference for 40 kHz ultrasonic cold extraction method
- >Line 152 0.22-um
- >Line 153 stored in dark? or stored in a non-transparent?
- >Line 156-159 please check on the configuration, it is not properly written.
- >Line 159 took place "in"
- >Line 160 information about the column should be written in the sentence instead of in parenthesis. Only the brand name and country should be in parenthesis.
- >Line 163 under the following "setting"
- >Line 164-165 the use of symbol is inconsistent, should it be C min-1 or C/min? also goes to Line 151.
- >Line 169 spectra per second? does it come with a symbol?
- >Line 170 expressed "in percentage of"...to "the" total
- >Line 172 authentic standards from where? any references? and what is Wiley? and please provide references and shortforms for these libraries.
- >Line 180 mentioned before, thus only shortform here
- >Line 182-183 using what program or technique?
- >Line 184 reference of software in parenthesis
- >Line 188 were first "identified using" the
- >Line 189-192 Figure 2A, 2B, Supplemental Figure S1 and S2 do not provide any useful information. suggest to remove.
- >Line 193 remove complete name for NIST as mentioned in Line 173
- >Line 193-194 60 compounds were tentatively identified with good match, which include Line 196 Supplemental Table S1 and Supplemental Figure S3 are useful and important information. They SHOULD NOT be supplemental data.
- >Line 196-197 this sentence should be in the MnM
- >Line 198 characterized by a high percentage of
- >Line 206 Figure 2C is irrelevant, it can be moved to the Supplementary data section.
- >Line 206-209 This should be in the Discussion section, not Results.
- >Line 211-212 This should be in the Discussion section, not Results.
- >Line 215 Figure 2D is irrelevant. The information can be presented in Supplementary Table S1
- >Line 218-220 This should be in the Discussion section, not Results.
- >Line 222 Figure 2E should be a standalone figure.
- >Line 222 I am not sure if this is correct, but how do you relate a heat map with a tree? should it be a group?
- >Line 223-227 This should not be in the Results.
- >Line 229-230 This should not be in the Results and replace "investigated" with "identified".
- >Line 231 how many agarwood samples? please state the number.
- >Line 233 Figure 3A should be replaced with a Table instead, with clear records of how many reads based on phylum, class, order, family and genus for all samples.
- >Line 234 with the Table proposed for Figure 3A, information in Figure 3B should be merged into the Table and Figure 3B would be irrelevant
- >Line 236 replaced "required" with "acquired"
- >Line 236 Figure 3C should be a standalone
- >Line 238 Figure 4A should be break-downed into Figure 4A-F. description of x-axis in simpson is missing.
- >Line 242-243 This sentence is not a Result, it is part of a Discussion
- >Line 243-244 This is MnM

>Line 245-248 This is Discussion, while Results for Figure 4B and 4C were not described. Figure 4B and 4C should not go under Figure 4. They should both go under a different Figure together. Same goes to Supplementary Figure S4.

>Line 250-251 I personally think that the heat-map is sufficient to show the results for the taxonomical analysis. The bar-chart is not relevant. Thus Supplemental Figure S6 is important and should be included in the manuscript.

>Line 254 Basidiomycota

>Line 267-269 This is Discussion

>Line 275-279 This is Discussion

>Line 282-286 This is Discussion

>Line 292 aromatic compounds

>Line 292 I do not understand why your correlation plot contain both fungal and chemical compounds in the same axis. Are you planning to see the correlation between fungal species and chemical compounds present in the sample? While this is the novelty of your study, yet it is in Supplemental form. You should choose to display the Supplemental Table S2 in your manuscript, and replace the yellow boxes with and underline or bold fonts to show the cv.

>Line 294 Figure 6 is irrelevant, while Supplemental Table S3 is an important result. However the size of the Table makes it difficult to be presented in the manuscript. In my opinion, it should be included in the text and the decimal points should be reduce to 3 digits; the correlation cv should be marked with underline or bold fonts instead of yellow highlights; the fungi names should be arranged in alphabetical order; The Y-axis should be the fungi names; The X-axis should be the compound numbers (with a general name "compounds" on top and only numbers below to reduce the use of "compound" for every number).

>Line 323 This is Discussion

>Line 325 This is not Result, instead it looks like a conclusion in the Discussion.

>Line 338 I do not think there is a distinct spatial-formed tissue in this study. There are all wood tissues.

>Line 296 A Discussion sentence in a Conclusion is not appropriate.

Reviewer: 2

Comments to the Author(s)

The present manuscript by Liu et al reports effects of fungal diversity on volatile compounds in different spatial formations of agarwood originated from *Aquilaria sinensis*. Authors have also showed relationship between fungal community and volatile compounds. The experiments are well conceived and well presented.

Major concerns:

- 1) where is reference of extraction method of agrawood oil? or Normally, the volatile oil is generally obtained from hrdrodistillation only. No discussion on extraction method relevant with extracting solvent provided in manuscript.
2. Use the common name for identified compounds. Kovat index is needed to present for identification confirmation. For example, mass spectrum of compounds 7 in Fig. 2 is not similar to those from library.
3. Fig S5B and Fig 5, other genus is the major group of fungi in all samples. The fungi in other genus may be correlated mainly to volatile compound production. The metagenome sequencing is important in this experiment. Genus of all isolated fungal strains are completely identified by PCR.
4. GCxGC chromatogram is not clear. Select one to present between 3D or 2D.

Author's Response to Decision Letter for (RSOS-190211.R0)

See Appendix A.

Decision letter (RSOS-190211.R1)

21-May-2019

Dear Dr Liu:

On behalf of the Editors, I am pleased to inform you that your Manuscript RSOS-190211.R1 entitled "Different wound locations of agarwood provide insights on the relevance between fungal diversity and volatile compounds in *Aquilaria sinensis*" has been accepted for publication in Royal Society Open Science subject to minor revision in accordance with the referee suggestions. Please find the referees' comments at the end of this email.

The Editors have identified that English language editing is required, please see <https://royalsociety.org/journals/authors/language-polishing/> for suggestions of services to use.

The reviewers and Subject Editor have recommended publication, but also suggest some minor revisions to your manuscript. Therefore, I invite you to respond to the comments and revise your manuscript.

- Ethics statement

- Data accessibility

If you wish to submit your supporting data or code to Dryad (<http://datadryad.org/>), or modify your current submission to dryad, please use the following link:
<http://datadryad.org/submit?journalID=RSOS&manu=RSOS-190211.R1>

- Competing interests

- Authors' contributions

- Acknowledgements

- Funding statement

Because the schedule for publication is very tight, it is a condition of publication that you submit the revised version of your manuscript before 30-May-2019. Please note that the revision deadline will expire at 00.00am on this date. If you do not think you will be able to meet this date please let me know immediately.

- 1) A text file of the manuscript (tex, txt, rtf, docx or doc), references, tables (including captions) and figure captions. Do not upload a PDF as your "Main Document".
- 2) A separate electronic file of each figure (EPS or print-quality PDF preferred (either format should be produced directly from original creation package), or original software format)

- 3) Included a 100 word media summary of your paper when requested at submission. Please ensure you have entered correct contact details (email, institution and telephone) in your user account
- 4) Included the raw data to support the claims made in your paper. You can either include your data as electronic supplementary material or upload to a repository and include the relevant doi within your manuscript
- 5) All supplementary materials accompanying an accepted article will be treated as in their final form. Note that the Royal Society will neither edit nor typeset supplementary material and it will be hosted as provided. Please ensure that the supplementary material includes the paper details where possible (authors, article title, journal name).

on behalf of Professor Xinguang Zhu (Associate Editor) and Kevin Padian (Subject Editor)
openscience@royalsociety.org

Associate Editor Comments to Author (Professor Xinguang Zhu):
Associate Editor
Comments to the Author:

I read the revision carefully. The revision effectively addressed the concerns by the reviewers and editors. The only concern now I have is that the language of the text needs to be improved. There are a number of grammatical errors in the text. Please have the whole manuscript either edited by an English editing service company or someone with proficient English skills.

Author's Response to Decision Letter for (RSOS-190211.R1)

See Appendix B.

Decision letter (RSOS-190211.R2)

04-Jun-2019

Dear Dr Liu,

I am pleased to inform you that your manuscript entitled "Agarwood wound locations provide insight into the association between fungal diversity and volatile compounds in *Aquilaria sinensis*" is now accepted for publication in Royal Society Open Science.

on behalf of Professor Xinguang Zhu (Associate Editor) and Kevin Padian (Subject Editor)
openscience@royalsociety.org

Associate Editor Comments to Author (Professor Xinguang Zhu):
Associate Editor: 1
Comments to the Author:
(There are no comments.)

Reviewer comments to Author:

Appendix A

Dear Professor Xinguang Zhu and Professor Kevin Padian:

On behalf of my co-authors, we thank you very much for giving us an opportunity to revise our manuscript, we appreciate editors and reviewers very much for their positive and constructive comments and suggestions on our manuscript entitled “Effects of fungal diversity on volatile compounds in different spatial formations of agarwood originated from *Aquilaria sinensis*”.

We have studied the editors’ reviewers’ comments carefully and have made revision which included a point by point response to the reviewers’ comments as following, and revised manuscript using 'Track Changes'/comments in Word. We have tried our best to revise our manuscript according to the comments. Attached please find the revised version, which we would like to submit for your kind consideration.

1. Associate Editor's comments (Professor Xinguang Zhu):

Dear Dr. Huang,

The submission has been reviewed by two domain experts. Based on their recommendations and also my evaluation for the manuscript, I judge that this paper requires a major revision before it can be accepted for publication in RSOS.

(1) I feel that the novelty of the paper requires clarification.

Yes, the editor is right. In the revised manuscript, we emphasized the novelty of this study according to the editor’s kind advice.

Our work was firstly reported that the different wounded points of fungal microbiome and agarwood chemical variety, which would provide insights on their intrinsic relevance in *A. sinensis*. Also, both high-throughput sequencing method to analyze the fungal community and GC×GC-HR-TOF-MS method to analyze the volatile compounds were successfully used in different agarwood samples formed from different wounded sections at the first time. We would

highlight the innovations from both scientific story and technical method.

(2) In addition, given that the observed correlation between the wood composition and also fungal diversity, authors suggest that this information can be used to guide future production of wood with altered wood quality, please describe in the text how this can be achieved in practice.

Yes, the editor is right. In the results and discussion sections of revised manuscript, we have added the above information and made a table (Table 4) including our results and relevant previous studies to give a guide of future production of agarwood.

(3) In addition, the authors correlated a few wood types with the fungal species and then make the statement that the fungal diversity influence the wood type. It is equally possible that the fungal species diversity was generated by the difference wood composition. Which is the cause, which is the result? Please discuss this point.

Yes, thank you for the review's kind suggestion.

In previous studies, researchers have recognized that the fungal community is highly specific to tree species as well as plant organ and tissue types (Cregger *et al.* 2018; Purahong *et al.* 2018). Thus, we choose the stems of *Aquilaria sinensis* as our experimental materials.

Naturally, agarwood is produced as a consequence of attack to microbes, insects and etc. Although both tissue types and wound methods could influence the agarwood types, we pay more attention to the effect of microbes on resin formation. Wood composition originated from various tissue types might influence the agarwood formation, but we pay more attention to the relation between microbes and agarwood compounds. Thus, in this article, we just emphasized the relation between the microbes and agarwood compositions through comparing the fungal communities and volatiles in various agarwood types. We hope the microbial fermentation could be a useful way to accumulate the agarwood formation or directly to obtain the agarwood resin ingredients in the future, which could effectively protect wild agarwood resources.

Cregger MA, Veach AM, Yang ZK, Crouch MJ, Vilgalys R, Tuskan GA, Schadt CW. 2018 The *Populus* holobiont: dissecting the effects of plant niches and genotype on the microbiome. *Microbiome* **6**, 31. (doi: 10.1186/s40168-018-0413-8)

Purahong W, Wubet T, Krüger D, Buscot F. 2018 Molecular evidence strongly supports deadwoodinhabiting fungi exhibiting unexpected tree species preferences in temperate forests. *ISME J.* **12**, 289-295. (doi: 10.1038/ismej.2017.177)

Please address all the other concerns of the reviewers as well. Looking forward to receiving a revised manuscript.

Best wishes,

Xinguang Zhu

SIPPE

2. Associate Editor: 2

Comments to the Author:

I have read this paper and found the results very interesting. I have some comments on this study.

(1) Please describe clearly the novelty of this paper.

Thanks for the editor's good advice. We have emphasized the novelty of this study in the revised manuscript from both scientific story and technical method as following. Our work was firstly reported that the spatial distribution of fungal microbiome and agarwood chemical variety, which would provide insights on their intrinsic relevance in *A. sinensis*. Also, both high-throughput sequencing method to analyze the fungal community and GC×GC-HR-TOF-MS method to analyze the volatile compounds were successfully used in different agarwood samples at the first time.

(2) Given that the observed correlation between the wood composition and also fungal diversity, authors suggest that this information can be used to guide future production of wood with altered wood quality. please describe in the text how this

can be achieved in practice.

Yes, thanks for the reviewer's good advice. In the results and discussion sections of revised manuscript, we have added the correlation between the wood composition and also fungal diversity and made a table (Table 4) including our results and relevant previous studies to give a guide of future production of agarwood.

(3) The authors correlated a few wood types with the fungal species and then make the statement that the fungal diversity influence the wood type. It is equally possible that the fungal species diversity was generated by the difference wood composition. Which is the cause, which is the result? Please discuss this point.

Yes, the reviewer is right. We have discuss the above issue in discussion part. Usually, the formation of natural agarwood is the result of tree defense reaction to microbial or insect attack. Although both tissue types and wound methods could influence the agarwood types, we pay more attention to the effect of microbes on resin formation. Wood composition originated from various tissue types might influence the agarwood formation, but we pay more attention to the relation between microbes and agarwood compounds. Thus, in this article, we just emphasized the relation between the microbes and agarwood compositions through comparing the fungal communities and volatiles in various agarwood types. We hope the microbial fermentation could be a useful way to accumulate the agarwood formation or to directly to obtain the agarwood resin ingredients in the future, which could effectively protect wild agarwood resources.

3. Reviewers' Comments to Author:

Reviewer: 1

Comments to the Author(s)

(1) The study reported on the fungal diversity and volatile compounds present at different parts of the *Aquilaria sinensis* tree upon wounding treatment. The chemical compounds obtained were correlated with the fungal diversity based on different tree parts. The latter work could be considered as novel to

agarwood-related studies. However, the manuscript was poorly presented and the authors should give more attention to their writings.

Thank for the reviewer's advice. We have totally revised our manuscript using 'Track Changes'/comments in Word. We discussed the correlation between volatiles and fungi in more detail in the revised manuscript. We hope our revision could meet the requirement of publication.

(2) there are several important information the author left out when reporting their work; the authors need to enhance their presentation by referring to published papers for the description in MnM; the authors tend to put Discussion materials in their Results; the authors need to determine which Table and Figure are important to display in the manuscript and which are not. I suggest that this manuscript can be accepted if the below comments are attained.

Yes, the reviewer is right. We have added more important published papers for the description in MnM in the reviewed paper. And we also rewrite the Discussion and Results to make the manuscript more readable and logical. We have also deleted Figure 2A, B, D and Supplemental Figure S1, S2, which do not provide any useful information. Also, we put Figure 2C into Supplemental materials, and Figure 3A, B were replaced with Table 1. Figure 4A were break-downed into Figure 4A-F. Figure 4B, C and Supplementary Figure S4 were both go under a different Figure together, new Figure 5. Supplemental Table S2 was displayed in manuscript as Table 3.

(3) Title. The authors only compared the fungal diversity and volatile compound between different tree parts, but the effects were not mentioned in the study.

Yes, the reviewer is right. According to the content of our manuscript, we would like to adjust the article title as "Different wounded locations of agarwood originated from *Aquilaria sinensis* provide insights on the intrinsic relevance between fungal diversity and volatile compounds", which meets the content of this work.

(4) Abstract. The abstract needs to be edited based on the comments below.

Thank you for the reviewer's advice. We have edited the abstract after

reviewing the manuscript, according to the comments below.

(5) Introduction. The authors did not emphasize the importance of fungal species identification in *Aquilaria* tree. There is no information regarding endophytic and introduced fungi in man-made agarwood induction process. While the Fungi-*Aquilaria* relationship is a crucial key to agarwood formation, the authors should provide more information on the importance of fungi towards agarwood formation in *A. sinensis*. And I believe the finding for this study is to contribute assistance towards sustainable agarwood production in cultivated *A. sinensis*, hence such information should also be included.

Thank you for the reviewer's kind suggestion. In the Introduction part, we have added more information associated with endophytic and introduced fungi in man-made agarwood induction process, which would help to analyze our data and give an assistance towards sustainable agarwood production in cultivated *A. sinensis*.

(6) MnM. Missing information on sampling techniques. Poor scientific write-up. The authors need to enhance their presentation by referring to published work for this section. I do not agree that the wounding treatment was carried out at different (spatial-formed) tissue types. The wounding points mentioned in the study were all from a single, or to breakdown, the most two tissue type: wood tissue from the stem and wood tissue from the branch. Scientifically, the five types of agarwood mentioned in this study are just names for the agarwood formed at different location on the tree stem. Instead, the authors should not mention the names of the agarwood in the MnM but to bring it up only at the Discussion section as an additional information on the naming practice among agarwood collectors in China. The samples, in my opinion, should be just treated as merely different wounding points.

Thank you for the reviewer's good advice. We have reviewed this part as reviewer suggested using 'Track Changes'/comments in Word. We have changed our title and related contents, for the five types of agarwood samples were formed from different locations of the stems not the tissue types. Additionally, the naming

practice of agarwood collectors were put into Discussion section not MnM. We also referred more published work to enhance our presentation, especially the sampling method.

(7) Results. The authors tend to put Discussion materials in the Results section, which is incorrect.

Yes, the reviewer is right. According to the reviewer's kind reminding, we have corrected the Results section, and put some discussing statement in Discussion section.

(8) Discussion. Poor discussed. With the amount of information obtained from the Results, the authors did not explain the possible causes of the strong correlation in certain fungi species and chemical compounds. There is possibility that the fungus is endophytic as well as introduced. furthermore, the samples were only collected at the end of the fifth year post-wounding treatment, this does not explain the correlation between the fungi and the agarwood formation because agarwood forms during the process. The process may have stopped at the 3rd or 4th year. My point is that to look at the true correlation between fungal diversity that would contribute towards agarwood formation, the authors should have carried out a time-lined sampling technique. Therefore, this study only explains the fungal diversity that is present when the agarwood has formed, and this does not directly provide information on the fungi whether they contribute towards the formation of agarwood in *A. sinensis*. This as well refers to the correlation between volatile compounds available and fungi diversity. In order to prove that the fungi are responsible to the great number of a certain volatile compound, there is a need to forward the study to specifically test on the efficacy of the fungus in contributing the production of a certain agarwood-related compound. Also, the authors should tone down their confidence between the volatile and fungi diversity recorded from their study as the study was only carried out in a single location. The fungal diversity could be affected by the environment, thus affect the result on the volatile compounds. Overall, the discussion needs to be refocused and rearranged to meet the standard of the study.

Thanks for the reviewer's kind advice. We have totally reviewed our discussion section. We hope that our reviewed discussion would meet the standard of the study.

(9) Figures & Tables

Removal and insertion of Figure and Tables are suggested in the comments below. The authors have to use good choice of language to present the Figure captions. If the data in the Figures or Tables were results from some software or database, there is a need to provide information of the said software or database. The Figures and Tables themselves should be able to be self-explanatory.

Thank you for the reviewer's good suggestion. We have removal and insertion of Figures and Tables according to the reviewer's suggestion. And we hope our revision could be more readable.

(10) Writing Problems.

Thanks very much for the reviewer's kind advices and revision. We have reviewed all the contents according to the reviewer's advices.

>Line 45-49 sentence is too long

Thanks for the reviewer's advice. We have divided the one sentence into two.

>Line 49 emesia? or amnesia?

Thanks for the reviewer's kind advice. Agarwood has been traditionally used as the antiasthma medicine in China and India. Please refer to references 8-10.

8. Wang S, Yu Z, Wang C, Wu C, Guo P, Wei J. 2018 Chemical constituents and pharmacological activity of agarwood and *Aquilaria* plants. *Molecules* **23**, 342. (doi: 10.3390/molecules23020342)

9. Liu JM, Gao YH, Xu HH, Xu ZQ. 2007 Chemical constituents of lignum *Aquilariae resinatum* (II). *Chinese Traditional Herbal Drugs* **38**, 1138-1140. (doi: 10.3321/j.issn:0253-2670.2007.08.004)

10. Miniyar PB, Chitre TS, Karve SS, Deuskar HJ, Jain KS. 2008 Antioxidant activity of ethyl acetate extract of *Aquilaria agallocha* on nitrite-induced methaemoglobin formation. *Int. J. Green Pharm.* **2**, 43-45. (doi: 10.4103/0973-8258.41185)

>Line 51 and "is" mainly

Do as reviewer requires.

>Line 51 *Aquilaria sinensis* (Lour.) Spreng.

Do as reviewer requires.

>Line 51 reference 11 and 12 are studies on *Aquilaria crassna*, not *A. sinensis*.

Thanks for the reviewer's kind reminding. We have changed the two references as following.

11. Xu Y, Zhang Z, Wang M, Wei J, Chen H, Gao Z, Sui C, Luo H, Zhang X, Yang Y, Meng H, Li W. 2013 Identification of genes related to agarwood formation: transcriptome analysis of healthy and wounded tissues of *Aquilaria sinensis*. *BMC Geomics* 14, 227. (doi: 10.1186/1471-2164-14-227)
12. Ye W, He X, Wu H, Wang L, Zhang W, Fan Y, Li H, Liu T, Gao X. 2018 Identification and characterization of a novel sesquiterpene synthase from *Aquilaria sinensis*: An important gene for agarwood formation. *Int J Biol Macromol* 108, 884-892. (doi: 10.1016/j.ijbiomac.2017.10.183)

>Line 55 International

Do as reviewer required.

>Line 56 "including *A. sinensis*" is redundant

Do as reviewer required.

>Line 57 "continuous use" is inappropriate, suggest to change it as "sustainable production"

Do as reviewer required.

>Line 58 "widely" cultivated

Do as reviewer required.

>Line 59 "Due to" its rarity

Do as reviewer required.

>Line 59-60 this is not a convincing statement. There are more than one species being rare and has crucial value that is widely cultivated, and why is only *A. sinensis* a good model?

Yes, the reviewer is right. According to the reviewer's advice, we changed the sentence as "Due to its rarity and crucial value, *A. sinensis* has been used to investigate processes for improving agarwood oil yield in China."

>Line 61, remove "usually"

Do as reviewer required.

>Line 62 consequence of attack to what?

In reference 13, the writer has reported the natural agarwood formation as following, "Naturally, the production of agarwood occurred due to microbial or insect attack and in the result, white heart wood converted into black/brown fragrant resin

which accumulates in main or side branch of stem.” Thus, in our article, we would like to analyze the relationship between the resin fragrance constituents and fungi using the agarwood samples which were only treated by simple wound method and then formed under natural conditions for many years imitating the natural agarwood formation.

Thanks for the reviewer’s advice. We have changed the sentence as “consequence of attack to microbes, insect and etc.”

13. Chhipa H, Chowdhary K, Kaushik N. 2017 Artificial production of agarwood oil in *Aquilaria sp.* by fungi: a review. *Phytochem. Rev.* **16**, 835-860. (doi: 10.1007/s11101-017-9492-6)

>Line 62 promotes

Do as reviewer required.

>Line 62 "Since" the last century

Do as reviewer required.

>Line 64 Fungi species such as

Do as reviewer required.

>Line 64 do not italic "sp."

Do as reviewer required.

>Line 69 reference 22 and 23 described the endophytic fungal diversity in different plant tissues (root, stem and leaf), while the study only involved the wood tissue in the stem and branch. Furthermore the fungal diversity reported in this study could also involved those being introduced from the surrounding due to the wounding treatment. Thus, the references provided here do not reflect as a support to this study.

Do as reviewer required. We have deleted these two references.

>Line 70 tissues of formation? incense? i do not understand what these mean.

>Line 70 citations?

Thanks for the reviewer’s kind reminding. We have changed this sentence as following, and added the citations (Chinese paper and book). “Due to different locations of formation, natural agarwood can be divided into various types in Chinese market [22, 23].”

22. Zhang F, Jiang Y. 2014 Agarwood collection entry encyclopedia. *Beijing, China: Chemical*

Industry Press.

23. Yang J, Mei W, Li W, Yu H, Zuo W, Dai H. 2015 TLC fingerprint of agarwood of *Aquilaria sinensis*. *J. Trop. Biol.* **6**, 189-196.

>Line 71-76 I do not think this information should be in the Introduction. instead it is part of MnM. the authors should briefly give some information on agarwood formation on different tree parts published by previous studies, giving the readers a hint on their hypothesis of the relationship between fungal diversity and different wounded parts.

Do as the reviewed required. We deleted this sentence, and rephrase the following sentence to give the readers a hint on their hypothesis of the relationship between fungal diversity and wounded part. "Although different types of agarwood have been widely traded in Chinese market, little work has been conducted aiming to holistically expound the relationship between variation in volatile compounds and fungal diversity."

>Line 76-79 please rephrase the sentence.

Do as the reviewed required. "Although different types of agarwood have been widely traded in Chinese market, little work has been conducted aiming to holistically expound the relationship between variation in volatile compounds and fungal diversity."

>Line 80-81 chemical characteristics "found in agarwood formed in different wounding points inflicted on the tree".

Thanks for the reviewer's advice. We have changed this sentence.

>Line 82 system "for" agarwood

Do as the reviewed required.

>Line 84 described

Do as the reviewed required.

>Line 91 structures found in different parts of agarwood formation

Do as the reviewed required.

>Line 92 When the authors mentioned to investigate the key fungi associated with different volatile compounds, I would be looking forward on a validation step where the fungi is isolated, then reintroduced back to the tree to confirm its strength

to produce those volatile, yet there is no such experiment carried out in this study. hence, i think the authors need to rephrase this objective.

Yes, the reviewer is right. We have changed this sentence as following: Furthermore, due to the regular pattern in fungal communities and the volatile constituents in different types of agarwood, we were able to characterize the fungal community structures found in different parts of agarwood formation and to preliminarily investigate the intrinsic relevance between fungal diversity and volatile compounds, which could help to furtherly understand the variety of agarwood fragrances.

>Line 96-98 what is the sampling technique? wounded using what? when was this carried out? how do you collect the formed agarwood? using what technique?

Thanks for the reviewer's advice. We have reviewed this part as following.

“30-year-old *A. sinensis* trees from Dalingshan town in Guangdong Province (latitude 22°45'43" N, longitude 113°48'45" E) were treated by physical wounding using machete [26,27]. Five years later, the agarwood section was formed about 0.5~1 cm thickness beneath the wounded surface, and were collected by cutting about 5 cm below the wounding section in November 2017, and then non-agarwood parts were removed.”

[26] Mohamed R. (2016) Agarwood: Science Behind the Fragrance. Berlin, Germany: Springer Press. doi: 10.1007/978-981-10-0833-7_3

[27] Tan CS, Isa NM, Ismail I, Zainal Z. 2019 Agarwood induction: current developments and future perspectives. Front. Plant Sci. 10, 122. (doi: 10.3389/fpls.2019.00122)

>Line 98-99 the number of individuals are unclear. each treatment has three individuals, so how many trees were there? how many treatments were there? is it 1 treatment 3 trees, then you got 5 different wounding points, so total is 15 tree? or u put all 5 wounding points on the same tree, then you have 3 biological replicates? Also, is there any negative control (a none wounded tree). How do you select your trees? are they side to side? or far apart? how trees interval? the information on the sampling technique is unclear. Please clarify.

Yes, it is 1 treatment 3 trees, 5 different wounding points, so total is 15 trees. This work only analyzed the agarwood samples formed from different locations of

trees, and we didn't analyze the samples from the none wounded tree. We selected the 30-year-old trees with similar diameter, about 20-30 cm, and the trees interval was about 5~20 meters.

>Line 99 the information in the Introduction should merge in here. However I have to raise my disagreement on the use of the name for the agarwood type as the specimen name, but to replace them by the organ/location of the tree where the wounding treatment is carried out as the specimen name. In such manner, the names for the type of agarwood form can be mentioned in the Discussion section. Unless the authors took the initiative to prepare an additional subsection in the MnM to specifically describe how the wounding treatment was carried out.

Yes, the editor is right. We have merged the information in the Introduction of sample collection here, and named the samples by number (I, II, III, IV, V) as following. The transverse section of the main trunk of *A. sinensis* trees were labeled as I (about 2.0-2.5 meter height); the transverse section of lateral branch of *A. sinensis* trees were labeled as II (about 1.5-1.8 meter height); the transverse section of the main trunk below the branches of *A. sinensis* trees were labeled as III (about 1.0-1.2 meter height); the longitudinal section of the main trunk of *A. sinensis* trees were labeled as IV (about 0.8-1.2 meter height); and the longitudinal section of the main trunk of *A. sinensis* trees near the root were labeled as V (about 0.2-0.4 meter height).

>Line 113 provide full name of ITS1, while its shortform in parenthesis. and i assume you amplify its sequence, not the whole gene.

>Line 114-115 provide complete sequences of the forward and reverse primers and their citation. and the reason why this primer set was chosen for this work was not mentioned.

Yes, the reviewer is right. We have reviewed this part as following: Internal transcribed spacer 1 (ITS1) sequences in all samples were amplified using the universal primers ITS5-1737F (5'-GGAAGTAAAAGTCGTAACAAGG-3') and ITS2-2043R (5'-GCTGCGTTCTTCATCGATGC-3') using a barcode to distinguish samples.

>Line 113-123 This whole part has to be rewrite. The way it is written is not

scientific. It is not mentioned that whose DNA was extracted and sequenced. please refer to published papers on how the methodology on the micro-array part should be presented. Machines should come with their brand name and country of make in parenthesis.

Do as reviewer required. We have totally reviewed this part carefully in manuscript.

“All agarwood samples were washed with distilled sterile water. Surfaces were sterilized by soaking the tissues sequentially in 75% ethanol for 2 min, followed by a rinse with sterile water, and dried with sterile filter paper. Then, DNA was extracted from 100 mg each agarwood sample using the MoBio PowerPlant Pro DNA Isolation Kit (MoBio Laboratories, Inc., Carlsbad, CA, USA). Agarwood samples had two replicate extractions per sample to achieve sufficient DNA yields. DNA purity was monitored on 1% agarose gels. All extractions were quantified on a NanoDrop 1000 spectrophotometer (NanoDrop Products, Wilmington, DE, USA). According to the concentration, DNA was diluted to 1 ng/μL using sterile water. Internal transcribed spacer 1 (ITS1) sequences in all samples were amplified using the universal primers ITS5-1737F (5'-GGAAGTAAAAGTCGTAACAAGG-3') and ITS2-2043R (5'-GCTGCGTTCTTCATCGATGC-3') using a barcode to distinguish samples. The PCR was performed with a Phusion[®] High-Fidelity PCR Master Mix (New England Biolabs, Ipswich, MA, UK). PCR was conducted according to Cregger *et al* [28]. PCR products were mixed in equal ratios. Then, the mixed PCR products were purified with a Qiagen Gel Extraction Kit (Qiagen, Germany). Sequencing libraries were generated using the Ion Plus Fragment Library Kit (Thermo Fisher Scientific, Waltham, MA, USA) following the manufacturer's recommendations, and index codes were added. Library quality was assessed on a Qubit[®] 2.0 Fluorometer (Thermo Fisher Scientific, Waltham, MA, USA) and Agilent Bioanalyzer 2100 system (Agilent Technologies Inc., CA, USA). The library was sequenced on a ThermoFisher Life Ion S5[™] platform (Thermo Fisher Scientific, Waltham, MA, USA). The high-throughput sequencing data are available at the database of National Center for Biotechnology Information (NCBI) in the Sequence Read Archive (SRA) database under BioProject number: PRJNA509099.”

>Line 127 built? please provide name of software.

Yes, the reviewer is right. We reviewed this sentence as following.

“Raw sequences were done with an Illumina MiSeq system (Illumina, San Diego, CA) by using MiSeq reagent kit v.3 (Illumina) to generate 2 × 300 bp reads at the University of Wisconsin Biotechnology Center Madison, WI, USA.”

>Line 128 cut into what size?

We reviewed “cut” as “removed”.

>Line 130 remove "under filtering conditions according to", replace with "using".

Do as reviewer required.

>Line 132 compared "against" the

Do as reviewer required.

>Line 132 reference for Unite database.

Do as reviewer required.

32. Nilsson RH, Larsson K-H, Taylor AFS, Bengtsson-Palme J, Jeppesen TS, Schigel D, Kennedy P, Picard K, Glöckner FO, Tedersoo L, Saar I, Kõljalg U, Abarenkov K. 2019 The UNITE database for molecular identification of fungi: handling dark taxa and parallel taxonomic classifications. *Nucleic Acids Res.* **47**, D259-D264. (doi: 10.1093/nar/gky1022)

>Line 133 "to" obtain

Do as reviewer required.

>Line 135 replace "by" with "using"

Do as reviewer required.

>Line 135 UPARSE, not Uparse.

Do as reviewer required.

>Line 136 reference for MOTHUR

Do as reviewer required.

35. Schloss PD, Westcott SL, Ryabin T, Hall JR, Hartmann M, Hollister EB, Lesniewski RA, Oakley BB, Parks DH, Robinson CJ, Sahl JW, Stres B, Thallinger GG, Van Horn DJ, Weber CF. 2009 Introducing mothur: open-source, platform-independent, community-supported software for describing and comparing microbial communities. *Appl. Environ. Microbiol.* **75**, 7537-7541. (doi: 10.1128/AEM.01541-09)

>Line 137 replace "reached the" with "achieved"

Do as reviewer required.

>Line 137-138 QIIME is a program name, remove the full name in parenthesis and replace it with its reference

Do as reviewer required.

36. Caporaso JG, Kuczynski J, Stombaugh J, Bittinger K, Bushman FD, Costello EK, Fierer N, Peña AG, Goodrich JK, Gordon JI, Huttley GA, Kelley ST, Knights D, Koenig JE, Ley RE, Lozupone CA, McDonald D, Muegge BD, Pirrung M, Reeder J, Sevinsky JR, Turnbaugh PJ, Walters WA, Widmann J, Yatsunenko T, Zaneveld J, Knight R. 2010 QIIME allows analysis of high-throughput community sequencing data. *Nat Methods* **7**, 335-336. (doi: 10.1038/nmeth.f.303)

>Line 139 reference for SILVA, Greengene and RDP database

Do as reviewer required.

37. Quast C, Priesse E, Yilmaz P, Gerken J, Schweer T, Yarza P, Peplies J, Glöckner FO. 2013 The SILVA ribosomal RNA gene database project: improved data processing and web-based tools. *Nucl. Acids Res.* **41**, D590-D596. (doi: 10.1093/nar/gks1219)

38. McDonald D, Price MN, Goodrich J, Nawrocki EP, DeSantis TZ, Probst A, Andersen GL, Knight R, Hugenholtz P. 2012 An improved Greengenes taxonomy with explicit ranks for ecological and evolutionary analyses of bacteria and archaea. *ISME J.* **6**, 610-618. (doi: 10.1038/ismej.2011.139)

39. Cole JR, Wang Q, Fish JA, Chai B, McGarrell DM, Sun Y, Brown CT, Porras-Alfaro A, Kuske CR, Tiedje JM. 2014 Ribosomal Database Project: data and tools for high throughput rRNA analysis. *Nucleic Acids Res.* **42**, D633-D642. (doi: 10.1093/nar/gkt1244)

>Line 139 replace "of" with "on"

Do as reviewer required.

>Line 140 this normalized data output. What this? please be specific.

Do as reviewer required. "Subsequent analyses on alpha and beta diversity were performed based on the normalized OTU abundance."

>Line 141 replace "Diagrams of Venn analysis" with "Venn diagrams"

Do as reviewer required.

>Line 141-142 some of the indexes and analyses come with their full name, please use their full names and put their shortnames in parenthesis

Do as reviewer required. We added the full names of the indexes and analyses of their full name as following.

"alpha diversity (including Chao1 and ACE richness estimators, Shannon and Simpson diversity indices, and phylogenetic diversity whole tree – PD_whole tree), beta diversity (including principal co-ordinates analysis – PCoA, principal

component analysis – PCA, and non-metric multidimensional scaling – NMDS)”

>Line 147 put the weight of the sample in the sentence, not in parenthesis.

Do as reviewer required.

>Line 147 2-mL

Do as reviewer required.

>Line 148 remove "Then"

Do as reviewer required.

>Line 150 reference for 40 kHz ultrasonic cold extraction method

Do as reviewer required. We added the reference of extraction method.

40. Liao G, Zhao M, Song X, Mei W, Yang J, Dai H. 2016 GC-MS Analysis of the agarwood produced by whole-tree agarwood-inducing technology. *Chin. J. Trop. Crops* 37, 411-417. (doi: 10.3969/j.issn.1000-2561.2016.02.031)

>Line 152 0.22-um

Do as reviewer required.

>Line 153 stored in dark? or stored in a non-transparent?

Do as reviewer required. "stored in a non-transparent glass bottle"

>Line 156-159 please check on the configuration, it is not properly written.

Do as the reviewer required.

"The GC×GC–HR-TOF-MS system consists of East & West 3300 GC×GC equipped with a TOF-MS (East & West Analytical Instrument Co., Beijing, China) used to acquire mass spectral data from the GC×GC."

>Line 159 took place "in"

Do as the reviewer required.

>Line 160 information about the column should be written in the sentence instead of in parenthesis. Only the brand name and country should be in parenthesis.

Yes, the reviewer is right. We have reviewed this sentence as following.

“The first separation took place in a conventional nonpolar GC column Agilent DB-5MS (25 m × 0.25 mm inner diameter × 0.25 μm film thickness), and the second was a medium GC column Agilent DB-17HT (3 m × 0.25 mm inner diameter × 0.15 μm film thickness).”

>Line 163 under the following "setting"

Do as the reviewer required.

>Line 164-165 the use of symbol is inconsistent, should it be C min-1 or C/min?
also goes to Line 151.

Do as the reviewer required. °C /min was changed into °C·min⁻¹.

>Line 169 spectra per second? does it come with a symbol?

Thanks for the reminding. "spectra per second" is usually without a symbol.

>Line 170 expressed "in percentage of"...to "the" total

Do as the reviewer required.

>Line 172 authentic standards from where? any references? and what is Wiley? and please provide references and shortforms for these libraries. >Line 180 mentioned before, thus only shortform here.

Do as reviewer required. We have reviewed this section as following.

“The identification of volatiles of the agarwood samples was based on an National Institute of Standards and Technology 11 (NIST11) library search combined with Kovats retention index (RI) [41]. The mass spectral match factor (probability>60) was used to judge whether a peak was correctly identified or not. For the determination of RI, calculated on the first dimension, a series of n-alkanes (C9 to C23) were used under the same experimental conditions. The computational formula of each compound RI was as follows:

$$RI = 100 \times \left\{ n + \frac{\lg t'(i) - \lg t'(n)}{\lg t'(n+1) - \lg t'(n)} \right\}$$

n and n+1 were the number of carbon atoms in alkanes eluting before and after the compound, respectively; t'(n) and t'(n+1) were the corresponding retention time and t'(i) was the retention time of the identified compound.”

41. Zhang N, Chen H, Sun B, Mao X, Zhang Y, Zhou Y. 2016 Comparative analysis of volatile composition in Chinese truffles via GC×GC/HR-TOF/MS and

electronic nose. *Int. J. Mol. Sci.* **17**, 412. (doi: 10.3390/ijms17040412)

>Line 182-183 using what program or technique?

Do as reviewer required. "Additionally, correlation analysis between fungal communities and volatile components was performed using the Pearson correlation method."

>Line 184 reference of software in parenthesis

Do as reviewer required.

42. SPSS Inc. 2007 SPSS for Windows, Version 16.0. Chicago, SPSS Inc. http://www.unimuenster.de/imperia/md/content/ziv/service/software/spss/handbuecher/englisch/spss_brief_guide_16.0.pdf

>Line 188 were first "identified using" the

Do as reviewer required.

>Line 189-192 Figure 2A, 2B, Supplemental Figure S1 and S2 do not provide any useful information. suggest to remove.

Do as reviewer required.

>Line 193 remove complete name for NIST as mentioned in Line 173

Do as reviewer required.

>Line 193-194 60 compounds were tentatively identified with good match, which include

Do as reviewer required.

>Line 196 Supplemental Table S1 and Supplemental Figure S3 are useful and important information. They SHOULD NOT be supplemental data.

Do as reviewer required.

>Line 196-197 this sentence should be in the MnM

Do as reviewer required.

>Line 198 characterized by a high percentage of

Do as reviewer required.

>Line 206 Figure 2C is irrelevant, it can be moved to the Supplementary data section.

Do as reviewer required.

>Line 206-209 This should be in the Discussion section, not Results.

Do as reviewer required.

>Line 211-212 This should be in the Discussion section, not Results.

Do as reviewer required.

>Line 215 Figure 2D is irrelevant. The information can be presented in Supplementary Table S1

Do as reviewer required.

>Line 218-220 This should be in the Discussion section, not Results.

Do as reviewer required.

>Line 222 Figure 2E should be a standalone figure.

Do as reviewer required.

>Line 222 I am not sure if this is correct, but how do you relate a heat map with a tree? should it be a group?

Heat map was used to show the relative content of five types of agarwood sample. The cluster tree was built according to the similarity of compound content, which showed the volatile difference between these samples.

>Line 223-227 This should not be in the Results.

Do as reviewer required.

>Line 229-230 This should not be in the Results and replace "investigated" with "identified".

Do as reviewer required.

>Line 231 how many agarwood samples? please state the number.

Do as reviewer required.

>Line 233 Figure 3A should be replaced with a Table instead, with clear records of how many reads based on phylum, class, order, family and genus for all samples.

Do as reviewer required. Figure 3A and B was replaced by Table 2.

>Line 234 with the Table proposed for Figure 3A, information in Figure 3B should be merged into the Table and Figure 3B would be irrelevant

Do as reviewer required. Figure 3A and B was replaced by Table 2.

>Line 236 replaced "required" with "acquired"

Do as reviewer required.

>Line 236 Figure 3C should be a standalone

Do as reviewer required. Figure 3C was replaced by figure 4.

>Line 238 Figure 4A should be break-downed into Figure 4A-F. description of x-axis in simpson is missing.

Do as reviewer required.

>Line 242-243 This sentence is not a Result, it is part of a Discussion

Do as reviewer required.

>Line 243-244 This is MnM

Do as reviewer required. We deleted this sentence.

>Line 245-248 This is Discussion, while Results for Figure 4B and 4C were not described. Figure 4B and 4C should not go under Figure 4. They should both go under a different Figure together. Same goes to Supplementary Figure S4.

Do as reviewer required. We make a new figure (figure 6), and described the results in figure 6.

>Line 250-251 I personally think that the heat-map is sufficient to show the results for the taxonomical analysis. The bar-chart is not relevant. Thus Supplemental Figure S6 is important and should be included in the manuscript.

Do as reviewer required.

>Line 254 Basidiomycota

Do as reviewer required.

>Line 267-269 This is Discussion

Do as reviewer required.

>Line 275-279 This is Discussion

Do as reviewer required.

>Line 282-286 This is Discussion

Do as reviewer required.

>Line 292 aromatic compounds

Do as reviewer required.

>Line 292 I do not understand why your correlation plot contain both fungal and chemical compounds in the same axis. Are you planning to see the correlation between fungal species and chemical compounds present in the sample? While this is the novelty of your study, yet it is in Supplemental form. You should choose to display the Supplemental Table S2 in your manuscript, and replace the yellow boxes with and underline or bold fonts to show the cv.

Thanks for the reviewer's good advice. We display the Supplemental Table S2 as reviewed manuscript Table 3.

>Line 294 Figure 6 is irrelevant, while Supplemental Table S3 is an important result. However the size of the Table makes it difficult to be presented in the manuscript. In my opinion, it should be included in the text and the decimal points should be reduce to 3 digits; the correlation cv should be marked with underline or bold fonts instead of yellow highlights; the fungi names should be arranged in alphabetical order; The Y-axis should be the fungi names; The X-axis should be the compound numbers (with a general name "compounds" on top and only numbers below to reduce the use of "compound" for every number).

Thanks for the reviewer's good advice. However, the Supplemental Table S3 was too large to make it in the text. But we put the extremely significant correlations ($p < 0.1$) between the fungal genera and compounds in the Table 4 and the whole data in Supplemental Table S1 in the reviewed paper.

>Line 323 This is Discussion

Do as reviewer required.

>Line 325 This is not Result, instead it looks like a conclusion in the Discussion.

Do as reviewer required.

>Line 338 I do not think there is a distinct spatial-formed tissue in this study. There are all wood tissues.

Yes, we have reviewed the title and related content in the reviewed paper.

>Line 396 A Discussion sentence in a Conclusion is not appropriate.

Do as reviewer required.

4. **Reviewer: 2**

Comments to the Author(s)

The present manuscript by Liu et al reports effects of fungal diversity on volatile compounds in different spatial formations of agarwood originated from *Aquilaria sinensis*. Authors have also showed relationship between fungal community and volatile compounds. The experiments are well conceived and well presented.

Major concerns:

(1) where is reference of extraction method of agarwood oil? or Normally, the volatile oil is generally obtained from hydrodistillation only. No discussion on extraction method relevant with extracting solvent provided in manuscript.

Thanks for the reviewer's kind advice. We have added the reference of extraction method of agarwood oil. Truly, the volatile oil is generally obtained from hydrodistillation. However, traditional extraction of volatile oil need a plenty of agarwood sample. For the agarwood samples are very precious, thus we used extracting solvent extraction method according to the related references.

(2) Use the common name for identified compounds. Kovat index is needed to present for identification confirmation. For example, mass spectrum of compounds 7 in Fig. 2 is not similar to those from library.

Thanks for the reviewer's kind advice. We have added the Kovat index to the compounds in Table 1 and the method was put in the section of Materials and Methods. All the mass spectrum of compounds were rechecked, and the mass spectrum of compounds 7 and 13 were put in Supplemental Figure S1.

(3) Fig S5B and Fig 5, other genus is the major group of fungi in all samples. The fungi in other genus may be correlated mainly to volatile compound production. The metagenome sequencing is important in this experiment. Genus of all isolated fungal strains are completely identified by PCR.

Yes, what the reviewer said is very reasonable. Although the fungi of other genus might be correlated to volatile compound, most of these fungi were quite different among one type of agarwood samples. If we analyzed these data, the

credibility of the results is not high. Thus, in this work, we only analyzed the major identified group of fungi, and we deleted Fig 5 and Fig S5B according to the reviewer 1.

(4) GCxGC chromatogram is not clear. Select one to present between 3D or 2D.

Thank you for the reviewer's advice. For there are many chemical compounds were shown in GCxGC chromatogram, which is not clearly enough. Thus, according to the suggestion from reviewer 1, we put Supplemental Table S2 in the reviewed manuscript as Table 1, and deleted all the chromatograms to make the data more useful and understandable.

We would like to express our great appreciation to you and reviewers for comments on our paper. Looking forward to hearing from you.

Thank you and best regards.

Yours sincerely,

Juan Liu

State Key Laboratory Breeding Base of Dao-di Herbs

National Resource Center for Chinese Materia Medica

Chinese Academy of Chinese Medical Sciences

No. 16, Nanxiaojie within Street, Dongcheng District, Beijing, China

Phone number: +86 15210752883

Email: juanliu126@126.com

CERTIFICATE OF ENGLISH EDITING

This document certifies that the paper listed below has been edited to ensure that the language is clear and free of errors. The edit was performed by professional editors at Editage, a division of Cactus Communications. The intent of the author's message was not altered in any way during the editing process. The quality of the edit has been guaranteed, with the assumption that our suggested changes have been accepted and have not been further altered without the knowledge of our editors.

TITLE OF THE PAPER

Agarwood wound locations provide insight into the association between fungal diversity and volatile compounds in *Aquilaria sinensis*

AUTHORS

Juan Liu, Xiang Zhang, Jian Yang, Junhui Zhou, Yuan Yuan, Chao Jiang, Xiulian Chi, Luqi Huang

JOB CODE

ZYJUA_7_2

Signature

Vikas Narang,
Senior Vice President,
Operations-Author Services, Editage

Date of Issue
June 02, 2019

Editage, a brand of Cactus Communications, offers professional English language editing and publication support services to authors engaged in over 500 areas of research. Through its community of experienced editors, which includes doctors, engineers, published scientists, and researchers with peer review experience, Editage has successfully helped authors get published in internationally reputed journals. Authors who work with Editage are guaranteed excellent language quality and timely delivery.

CACTUS

Contact Editage

Worldwide
request@editage.com
+1 877-334-8243
www.editage.com

Japan
submissions@editage.com
+81 03-6868-3348
www.editage.jp

Korea
submit-
korea@editage.com
1544-9241
www.editage.co.kr

China
fabiao@editage.cn
400-005-6055
www.editage.cn

Brazil
contato@editage.com
0800-892-20-97
www.editage.com.br